# On-surface cyclization of vinyl groups on poly-para-phenylene involving an unusual pentagon to hexagon transformation

Marco Di Giovannantonio [1,2,9] ✉, Zijie Qiu [3,4,9], Carlo A. Pignedoli [1,9], Sobi Asako [5,6], Pascal Ruffieux [1], Klaus Müllen [3,7] ✉, Akimitsu Narita [3,6] ✉ & Roman Fasel [1,8] ✉

On-surface synthesis relies on carefully designed molecular precursors that are thermally activated to afford desired, covalently coupled architectures. Here, we study the intramolecular reactions of vinyl groups in a poly-*para*-phenylene-based model system and provide a comprehensive description of the reaction steps taking place on the Au(111) surface under ultrahigh vacuum conditions. We find that vinyl groups successfully cyclize with the phenylene rings in the *ortho* positions, forming a dimethyl-dihydroindenofluorene as the repeating unit, which can be further dehydrogenated to a dimethylene-dihydroindenofluorene structure. Interestingly, the obtained polymer can be transformed cleanly into thermodynamically stable polybenzo[*k*]tetraphene at higher temperature, involving a previously elusive pentagon-to-hexagon transformation via ring opening and rearrangement on a metal surface. Our insights into the reaction cascade unveil fundamental chemical processes involving vinyl groups on surfaces. Because the formation of specific products is highly temperature-dependent, this innovative approach offers a valuable tool for fabricating complex, low-dimensional nanostructures with high precision and yield.

On-surface synthesis is a highly effective bottom-up method for growing one- and two-dimensional (1D and 2D) nanostructures, enabling precise control over their shapes and compositions at the atomic level[1,2]. To visualize and thoroughly characterize the resulting compounds, advanced microscopy techniques and theoretical simulations are employed. Notably, various counterparts of well-known organic reactions, including Ullmann coupling[3,4], Glaser coupling[5–7], Masamune–Bergman cyclization[8,9], and Diels–Alder cycloaddition[10,11],

have been successfully executed on metal surfaces under ultrahigh vacuum (UHV) conditions. Additionally, recent years have witnessed the development of unique on-surface processes that are not easily accessible through traditional solution chemistry, such as the aliphatic C–H selective activation[12,13]. Despite these significant advancements, there remains untapped potential in exploiting the available toolkit for on-surface synthesis, which, in some cases, hampers the realization of complex architectures.

[1]Empa, Swiss Federal Laboratories for Materials Science and Technology, nanotech@surfaces Laboratory, 8600 Dübendorf, Switzerland. [2]Istituto di Struttura della Materia – CNR (ISM-CNR), 00133 Roma, Italy. [3]Max Planck Institute for Polymer Research, 55128 Mainz, Germany. [4]School of Science and Engineering, Shenzhen Institute of Aggregate Science and Technology, The Chinese University of Hong Kong, Shenzhen (CUHK-Shenzhen), Guangdong 518172, P.R. China. [5]RIKEN Center for Sustainable Resource Science, Wako, Saitama 351-0198, Japan. [6]Organic and Carbon Nanomaterials Unit, Okinawa Institute of Science and Technology Graduate University, Okinawa 904-0495, Japan. [7]Department of Chemistry, Johannes Gutenberg University Mainz, Duesbergweg 10-14, 55128 Mainz, Germany. [8]Department of Chemistry, Biochemistry and Pharmaceutical Sciences, University of Bern, 3012 Bern, Switzerland. [9]These authors contributed equally: Marco Di Giovannantonio, Zijie Qiu, Carlo A. Pignedoli. ✉e-mail: marco.digiovannantonio@ism.cnr.it; muellen@mpip-mainz.mpg.de; akimitsu.narita@oist.jp; roman.fasel@empa.ch

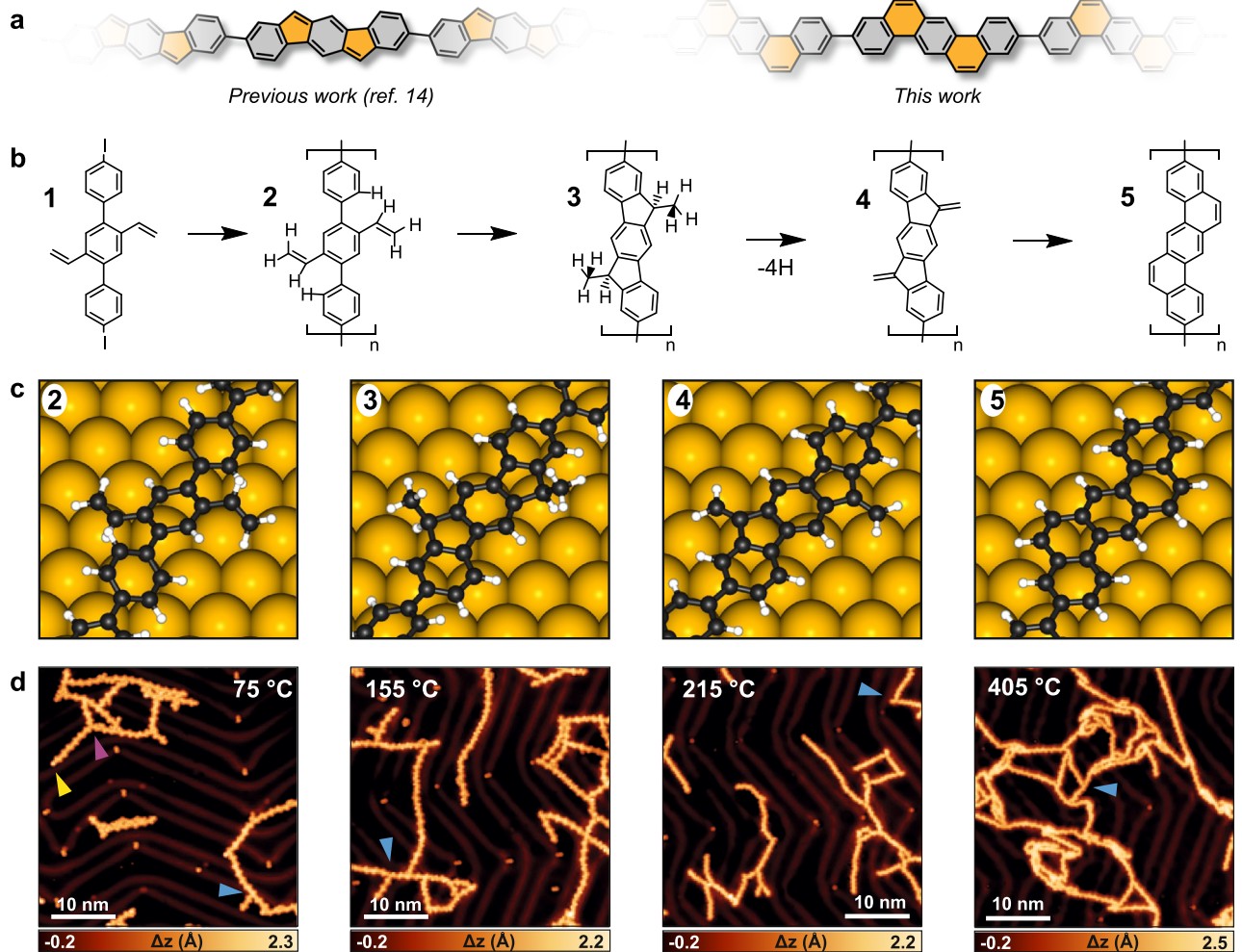

**Fig. 1 | On-surface reactivity of vinyl groups. a** Indenofluorene or benzo[*k*]tetraphene scaffolds in carbon-based nanostructures can be obtained using methyl or vinyl groups. **b** Schematics of the vinyl reaction pathway observed in this work. **c** DFT-optimized adsorption geometries of oligomers of structures **2**–**5**, made of three repeating units. **d** Large-scale STM images at relevant reaction steps.

Colored arrows indicate the side products described in the text. Images were acquired at 4.7 K after annealing the sample at the indicated temperature. Scanning parameters: $I_t = 30$ pA, $V_b = -1.0$ V; $I_t = 30$ pA, $V_b = -0.3$ V; $I_t = 30$ pA, $V_b = -0.3$ V; $I_t = 100$ pA, $V_b = -0.1$ V.

Recently, we reported on the successful oxidative cyclization of methyl groups anchored on the armchair edge of a polyphenylene chain on the Au(111) surface, yielding five-membered rings at desired positions (Fig. 1a)[14,15]. While similar strategies have been employed to create six- and seven-membered rings in nanographenes using methyl groups at different positions[16–18], these structures are prone to thermal detachment[19]. In graphene nanoribbons (GNRs), the extension of an armchair edge often necessitates the inclusion of additional six-membered rings. For instance, by extending the edge of 7-atom-wide armchair GNRs (7-AGNRs), it becomes possible to obtain 8-AGNRs, which are predicted to possess notably narrow bandgaps[20]. However, achieving precise formation of 8-AGNRs on surfaces using a simple monomer has proven challenging[21]. On-surface methyl–methyl coupling has been explored to extend the armchair edge or bay region of nanographenes and was successful for circumcoronene[22], but failed to yield uniformly edge-extended 9-AGNRs[23]. Considering the success in constructing ladder-type nanostructures by olefin metathesis in solution[24], the use of vinyl groups could potentially offer an elegant approach to the formation of six-membered rings. To the best of our knowledge, however, such an approach has not been described thus far.

In solution synthesis, cyclization reactions of alkenes and alkynes are powerful methods to extend π-conjugation and create polycyclic aromatic and heteroaromatic compounds[25–27]. The outcome of such intramolecular cyclization reactions can be divided into *exo*- and *endo*-types, whose kinetically favored products have been discussed as Baldwin's rules[28–30]. For example, depending on the substrate and reaction conditions (e.g., catalyzed/mediated by different transition metals or Lewis acids), the annulation of *ortho*-alkynylbiaryls can produce phenanthrene frameworks via a 6-*endo-dig* ring closure, or yield fluorene derivatives through a 5-*exo-dig* cyclization[31]. For *ortho*-alkenylbiaryls, the hypervalent iodine-mediated and Pd-catalyzed intramolecular oxidative cyclizations were reported to yield phenanthrene derivatives[32] and alkylidene fluorenes[33], respectively. The related cyclization reactions are summarized in the Supplementary Fig. 2. However, the selectivity of intramolecular cyclization reactions towards 6-*endo*- and 5-*exo*-products has never been investigated in on-surface chemistry.

In this work, the on-surface intramolecular cyclization of vinyl groups with *ortho* phenylene rings was investigated on an atomically flat Au(111) surface under UHV conditions. We used low-temperature scanning tunneling microscopy (STM) and noncontact atomic force microscopy (nc-AFM) with CO functionalized tips to identify the chemical structures of the precursor, intermediates, and final product. Poly(2′,5′-divinyl-*p*-terphenylene) (**2**) was obtained by initial dehalogenation aryl–aryl coupling of a 4,4″-diiodo-2′,5′-divinyl-1,1′:4′,1″-terphenylene precursor (**1**) at 75 °C. Notably, during the annealing process, poly(-dimethyl-dihydroindenofluorene) intermediate **3** was identified as the

5-*exo-trig* cyclized product, supported by experimental data and density functional theory (DFT) calculations. Further heating to 215 °C induced the dehydrogenation of **3** and afforded exclusively poly(dimethylene-dihydroindenofluorene) **4**, which was then completely transformed into polybenzo[*k*]tetraphene **5** upon annealing at 405 °C. The transformation of five-membered rings in **4** into hexagonal structures in **5** is accompanied by an energy gain of 1.2 eV per repeating unit. Our detailed report of the on-surface reaction pathway of vinyl groups and *ortho* phenylene rings provides invaluable insights for future synthetic approaches towards ladder-type nanostructures.

## Results

To explore the possibility of using vinyl groups for edge extension of carbon nanomaterials on metal surfaces, we synthesized precursor **1** (Supplementary Fig. 1) aiming at the formation of 1D poly(*p*-phenylene) decorated with vinyl functionalities as a model. Iodine was selected as the halogen substituent with the lowest dissociation energy[34] to activate the dehalogenative aryl–aryl coupling while avoiding the coupling between vinyl groups and halogenated sites (Heck reaction)[35]. After deposition of **1** on the Au(111) surface held at room temperature (RT) in UHV, intact precursor molecules were observed, either isolated or packed into islands (Fig. 2 and Supplementary Fig. 3). A few molecules already formed dimers and trimers, indicating that deiodination was initiated already at RT (Supplementary Fig. 3). C–I bond dissociation on Au(111) typically occurs between RT and 80 °C, and depends on the molecular structure and conformation[36–40]. Hence, it is not surprising that 20% of C–I bonds (from 140 molecules analyzed) already dissociated upon RT deposition. However, the majority of the molecules were still intact. Constant-height frequency shift nc-AFM images with a CO functionalized tip allowed the identification of the chemical structure of an isolated molecule (Fig. 2b, c). The bright protrusions at the two ends of each unit are attributed to the iodine atoms connected to the molecular backbone, while the two diagonal features intersecting the molecular axis can be assigned to the vinyl groups (Fig. 2a). Due to the steric repulsion between hydrogen atoms, the three benzene units in the *p*-terphenyl backbone are tilted with respect to each other. As a result, constant-height nc-AFM images reveal only one of the two vinyl groups (labeled as "vinyl up" in Fig. 2d), together with the opposite side of the two external benzene units. Here, the four hydrogen atoms pointing away from the surface produce the main contrast in the nc-AFM image.

After mild annealing of this sample to 75 °C, most of the molecules were found to form 1D poly(2',5'-divinyl-*p*-terphenylene) (**2**) (Supplementary Fig. 3). High-resolution STM imaging of a polymer segment shows diagonal features (Fig. 2g), similar to those present in precursor **1**. nc-AFM imaging of this polymer (Fig. 2h) reveals an alternating sequence of features ascribed to vinyl groups and tilted benzene units (guide scheme in Fig. 2j), as in precursor **1**. This observation confirms that polymer **2**, bearing unreacted vinyl groups, was obtained. At this stage of the reaction, almost 90% of the active sites (carbon radicals after deiodination) have reacted according to the targeted scheme and afforded the expected C–C bonds (Fig. 1b). The STM image in Fig. 1d additionally revealed a minority of side products. T-shaped connections are also observed (7%), due to a limited, yet unavoidable, Heck-type reaction between carbon radicals and vinyl groups (cyan arrows in Fig. 1d). Iodine atoms are still connected to 2% of the sites (yellow arrow in Fig. 1d), due to the moderate annealing temperature, and some triple connections (2%) are also observed (magenta arrow in Fig. 1d). Such composition is similarly reflected in the samples annealed at higher temperatures (Fig. 1d), with the T-shaped connections occurring in less than 10% of the active sites, and the iodinated sites and triple connections absent, leaving more than 90% of the targeted intermolecular connections. The fate of iodine atoms in the present reaction does not differ from what is typically observed in other on-surface synthesis studies based on deiodination processes[37,41]. This

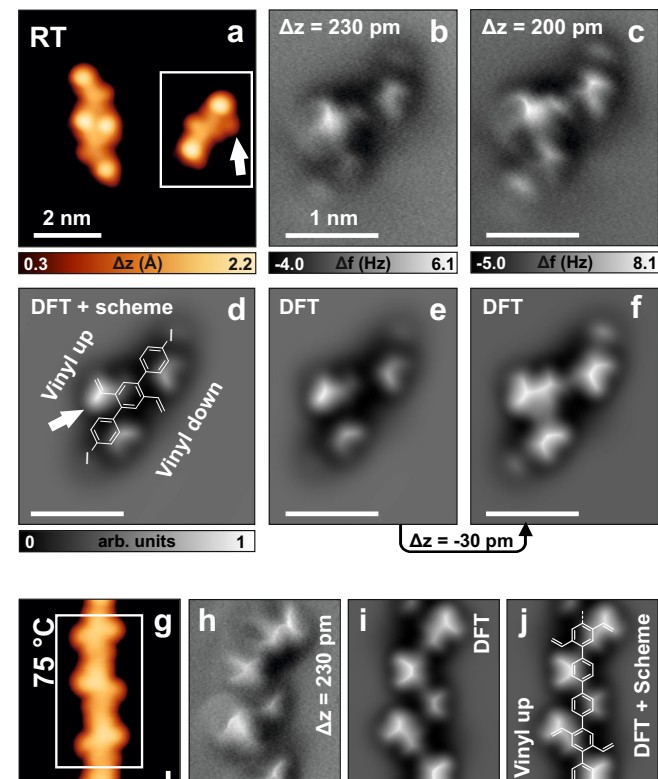

**Fig. 2 | Precursor 1 and poly(2',5'-divinyl-*p*-terphenylene) (2). a, g** STM images of intact molecules and polymers obtained after deposition of **1** on Au(111) held at RT and subsequent annealing to 75 °C. The white arrow in (**a**) indicates an adsorbate next to the isolated molecule. **b, c** Constant-height frequency shift nc-AFM images with a CO functionalized tip of an isolated molecule **1** (highlighted in (**a**) with a white rectangle) at decreasing tip-surface distance. **d–f** Simulated nc-AFM images of an intact molecule **1** on Au(111) at different tip-sample distances, with and without the molecular scheme superimposed. The white arrow in (**d**) highlights the bright feature due to the hydrogen protruding from the α-carbon of the vinyl group. Due to the rotation of the phenylene rings, one vinyl group points towards the substrate and is therefore not clearly imaged by nc-AFM. **h** nc-AFM image of a segment of **2** (indicated in (**g**) with a white rectangle). **i, j** Simulated nc-AFM images of a polymer strand, without and with molecular scheme superimposed. Scanning parameters: **a** $I_t = 30$ pA, $V_b = -0.2$ V; **b, c, h** $\Delta z$ above the STM set point ($I_t = 100$ pA, $V_b = -5$ mV) is given for each image; **g** $I_t = 50$ pA, $V_b = -0.3$ V. Temperatures in (**a** and **g**) indicate at which temperature the samples were annealed before cooling down to 4.7 K for imaging.

includes chemisorption on the metal substrate after dissociation from the molecular precursors, with the iodine preferentially adsorbed at the elbow sites of the herringbone reconstruction of the Au(111) surface, and next or in between polymer chains. At more elevated temperatures, iodine desorbs (as HI, after recombination with hydrogen released in the next reaction steps, *vide infra*).

After annealing to 155 °C, roughly 30% of the vinyl groups transformed into bright dot-like features (Fig. 3a, b), while the remaining 70% still featured unreacted vinyl groups. We performed nc-AFM imaging of a polymer segment hosting two such dots surrounded by unreacted vinyl groups. The newly formed moieties are significantly higher than pristine vinyl groups with respect to the polymer plane, and the two dots dominate the image (Fig. 3c, d). To identify the chemical structure of these sites, we simulated the nc-AFM image of various DFT-optimized

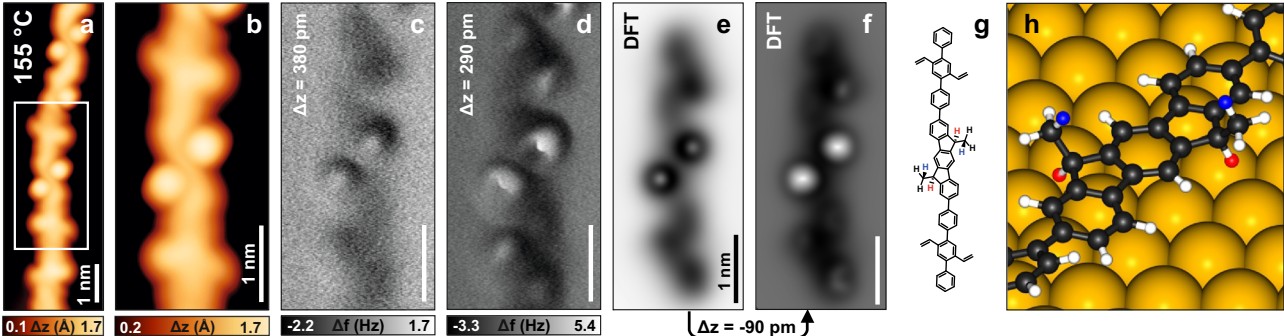

**Fig. 3 | Methyl fluorene intermediate (3). a** STM image of a polymer segment after annealing the sample to 155 °C and **b** zoom in of the highlighted part. **c, d** Constant-height frequency shift nc-AFM images with a CO functionalized tip of the segment in (**b**) at decreasing tip-surface distance. **e, f** Simulated nc-AFM images of the DFT-optimized structure in (**g**) on Au(111). **g** Red and blue hydrogen atoms point toward and away from the surface, respectively. **h** Highlight of the central unit of the structure in (**g**). Scanning parameters: **a, b** $I_t = 30$ pA, $V_b = -0.3$ V; **c, d** $\Delta z$ above the STM set point ($I_t = 100$ pA, $V_b = -5$ mV) is given for each image. Temperature in (**a**) indicates the sample annealing temperature, while subsequent imaging was performed at 4.7 K.

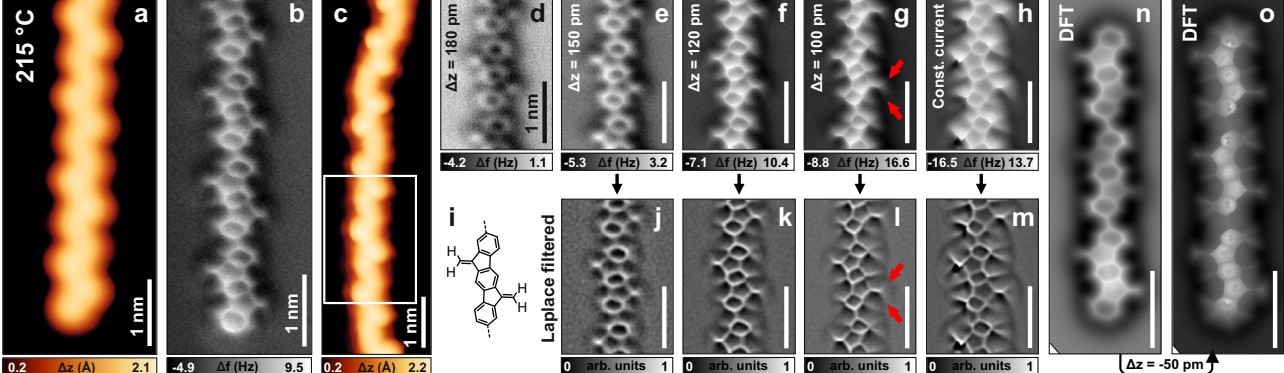

**Fig. 4 | Dimethylene-dihydroindenofluorene intermediate 4. a, c** STM images of two polymer segments after annealing the sample to 215 °C, the former displaying a chain termination. **b, d–g** Constant-height frequency shift nc-AFM images with a CO functionalized tip of the segments in (**a**) and the highlighted part in (**c**), respectively. **h** Constant-current frequency shift nc-AFM image with a CO functionalized tip of the segment highlighted in (**c**). **i** Molecular scheme of a repeating unit of **4**. **j–m**, Laplace-filtered images. **n, o** Simulated nc-AFM images of a trimer **4**. Scanning parameters: **a** $I_t = 100$ pA, $V_b = -0.1$ V; **b** $\Delta z$ is 160 pm above the STM set point ($I_t = 100$ pA, $V_b = -5$ mV). **c** $I_t = 200$ pA, $V_b = -20$ mV; **d, g** $\Delta z$ above the STM set point ($I_t = 100$ pA, $V_b = -5$ mV) is given for each image. **h** $I_t = 150$ pA, $V_b = -5$ mV. Temperature in (**a**) indicates the sample annealing temperature, imaging was performed at 4.7 K.

geometries (Supplementary Fig. 4). All these geometries featured a cyclization into a five-membered ring, based on our experimental findings after annealing at a higher temperature (*vide infra*). Among the simulated structures, the one that best reproduced our experimental images is reported in Fig. 3g, h. The agreement between the simulated nc-AFM images of such structure (Fig. 3e, f) and the experimental data (Fig. 3c, d) confirms that the two dot-like features originate from the dimethyl-dihydroindenofluorene structure. This intermediate structure **3** is obtained upon cyclization of the vinyl groups to the neighboring phenylene rings, forming bonds to the α-carbons and thus the 5-*exo-trig* product with a five-membered ring. A concomitant hydrogen migration also happens, leading to a sp³ hybridized five-membered ring and a methyl group protruding away from the surface.

Further annealing of the sample to 215 °C produced the complete disappearance of both the diagonal and the dot-like features, in favor of polymers with a homogeneous zigzag shape (Fig. 4a). nc-AFM imaging of these chains revealed the presence of five-membered rings at the opposite sides of each repeating unit (Fig. 4b). The peculiar appearance of their apexes deserved careful investigation; therefore, we acquired height-dependent constant-height nc-AFM images of one polymer segment (Fig. 4d–g). At decreased tip-sample distance, the region external to the apex of each five-membered ring appeared with a Y shape, due to an elongated segment protruding out of the

apex that divided into two weaker features (red arrows in Fig. 4g). This picture is more evident in the constant-current nc-AFM images (Fig. 4h), as well as the corresponding Laplace-filtered images (Fig. 4j–m). These protruding features are attributed to methylenes substituted on the apexes of fluorene substructures, with the central segment being the C = C unit, and the two weaker lobes being the hydrogen atoms of the CH₂ moiety. Hence, the chemical structure of this polymer can be identified as **4** in Fig. 4i. The nc-AFM simulations of a DFT-optimized trimer of **4** on Au(111) at different scan heights are reported in Fig. 4n, o, showing good agreement with the experimental images.

Our experimental results demonstrate that polymer structure **4** is formed on Au(111) after annealing to 215 °C. To achieve the desired armchair edge extension with six-membered rings, we annealed **4** to temperatures beyond 215 °C. Structure **4** is very stable and is the only observed product until 380 °C. When annealed at 380 °C, although STM images do not show substantial differences (Fig. 5a), nc-AFM images of the polymers reveal that some repeating units underwent ring rearrangement (Fig. 5b, c). Thereby, some of the fluorene substructures were converted into phenanthrene, transforming dimethylene-dihydroindenofluorene as in polymer **4** into benzo[*k*]tetraphene as in polymer **5** (blue and red arrows in Fig. 5b, respectively). This rearrangement was sometimes observed for only one of the two pentagonal

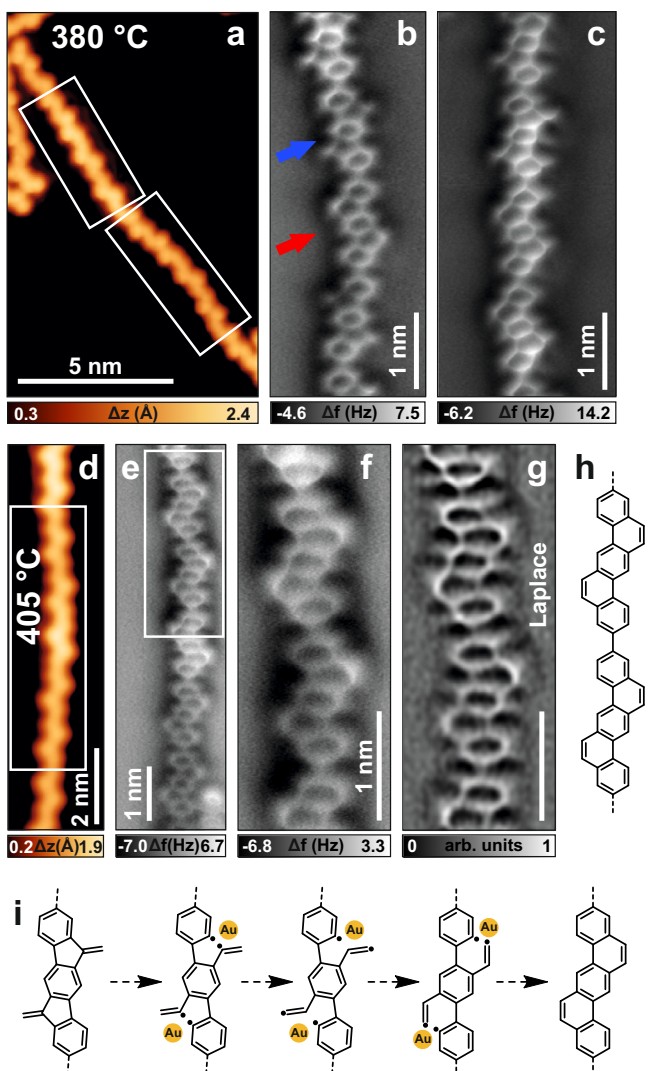

**Fig. 5 | Final conversion to polybenzo[k]tetraphene 5. a, d** STM images of two polymer segments after annealing the sample to 380 and 405 °C, respectively. **b, c, e, f** Constant-height frequency shift nc-AFM images with a CO functionalized tip of the segments in (**a** and **d**). **g** Laplace-filtered image of (**f**). **h** Molecular scheme of **5. i** Sketch of the ring rearrangement mechanism, mediated by gold adatoms. Scanning parameters: **a** $I_t$ = 100 pA, $V_b$ = –20 mV; **b, c** Δz is 170 and 130 pm above the STM set point ($I_t$ = 100 pA, $V_b$ = –5 mV). **d,** $I_t$ = 100 pA, $V_b$ = –0.1 V; **e, f** Δz is 160 and 150 pm above the STM set point ($I_t$ = 100 pA, $V_b$ = –5 mV). Temperatures in (**a** and **d**) indicate the sample annealing temperature, while imaging was performed at 4.7 K.

sites of a repeating unit (Supplementary Fig. 5), indicating that the transformation happened independently. From the available nc-AFM images, we could estimate that 35% of the five-membered rings converted into six-membered rings at 380 °C. At this temperature, we noted the appearance of some covalent couplings between adjacent chains, as minority side products. Such cross-linking is due to a C–H activation of the methylene groups (see, e.g., Supplementary Fig. 5). To complete the pentagon-to-hexagon conversion, the sample was further annealed to a slightly higher temperature. To our delight, at 405 °C the desired polybenzo[k]tetraphene **5** was achieved efficiently as the main final product (Fig. 5d–h).

The clearly resolved polymer structures (**2, 4**, and **5**) and the clean pentagon-to-hexagon transformation are remarkable, prompting us to investigate the underlying mechanism and reaction sequence. To this purpose, we first focused on the initial vinyl cyclization and modeled the two competing processes of pentagon and hexagon formation, i.e., *5-*

*exo-trig* and *6-endo-trig*, respectively (Fig. 6). To promote the cyclization into a five-membered ring, the vinyl group should take a conformation where the $CH_2$ group is pointing away from the polymer backbone, so that the α-carbon of the vinyl is closer to the phenylene ring in the *ortho* position, compared to the β-carbon. We optimized the geometry of a single repeating unit of polymer **2** in such conformation on Au(111) by DFT and used its energy as a reference (model structure **II** at 0 eV in Fig. 6). On the other hand, to promote the cyclization into a six-membered ring, the vinyl group should rotate so that the β-carbon comes closer to the phenylene ring. This conformation appears higher in energy than the previous one by 0.42 eV. Therefore, during the annealing of the sample, a vinyl group should prefer the former conformation, which renders the five-membered ring formation more likely. Moreover, to create a C–C bond in the case of pentagon formation, the reaction undergoes a transition state that is more favored (by 0.39 eV) than in the hexagon formation (Fig. 6). As cyclization reactions can be affected by the presence of metal adatoms next to the reacting site[42], we additionally modeled the same scenario including the presence of a gold adatom in proximity of the vinyl group, and found the same qualitative preference for the five-membered ring formation (Supplementary Fig. 6). These observations support the experimental evidence of an exclusive on-surface *5-exo-trig* cyclization of the vinyl groups, although the *6-endo-trig* cyclization would also be permitted according to Baldwin's rules.

After the migration of four hydrogen atoms per molecular unit (not computed here), model structure **III** exhibits a very stable geometry (–2.05 eV). The formation of **IV** requires the removal of four hydrogen atoms per repeating unit of **III**. According to the DFT calculations, the activation energy for the removal of hydrogen attached to the apex of the five-membered ring and from the methyl group of **III** is 1.26 eV and 1.21 eV, respectively (see Fig. 6). Polymer **4** is very stable and experimentally observed over a wide temperature range. Nevertheless, DFT calculations suggest that the total energy per unit **IV** is still 1.24 eV higher than that of **V** on Au(111) (Fig. 6). The achievement of the thermodynamically stable product **V** requires the opening of the five-membered ring via C–C bond cleavage and formation of another C–C bond accompanying hydrogen migration (Fig. 5i). An in-depth investigation of such complicated mechanism would require advanced techniques such as metadynamics[43], which is computationally prohibitive within DFT for our system and beyond the scope of the present study. Here, we aim at offering a plausible description of the initial ring opening without computing the entire process. We focus on the C–C bond rupture, which could represent one of the possible initiation steps to transform **IV** into **V**. Other possible paths, such as initial hydrogen migration, are not considered here and are out of the scope of the present work. DFT modeling of the initial C–C bond breaking resulted in an energy barrier larger than 3 eV, which was reduced to 2.30 eV if a gold adatom was placed next to the targeted C–C bond (Supplementary Fig. 8). This suggests that the pentagon-to-hexagon transformation, although experimentally requiring much higher temperature than the previous reaction steps, could be facilitated significantly by surface adatoms, which are not available in traditional solution chemistry. The calculated energy barrier of 2.30 eV with adatoms is high enough to justify the stability of polymer **4** up to 380 °C. It is thus possible to obtain the targeted ring topology – 5- or 6-membered rings as in polymer **4** and **5**, respectively – with high yield by adjusting the annealing temperature. Besides, the gold adatom can stabilize the diradical intermediate after C–C bond cleavage (Fig. 5i and Supplementary Fig. 8), facilitating the successive hydrogen migration and final C–C bond formation.

## Discussion

In conclusion, we have provided detailed insight into the on-surface cyclization of vinyl groups to neighboring phenylene rings. A model terphenyl precursor **1** was synthesized to grow vinyl-decorated polyphenylene **2** and to monitor the thermally activated intramolecular reactions of vinyl groups on a Au(111) surface. High-resolution nc-AFM

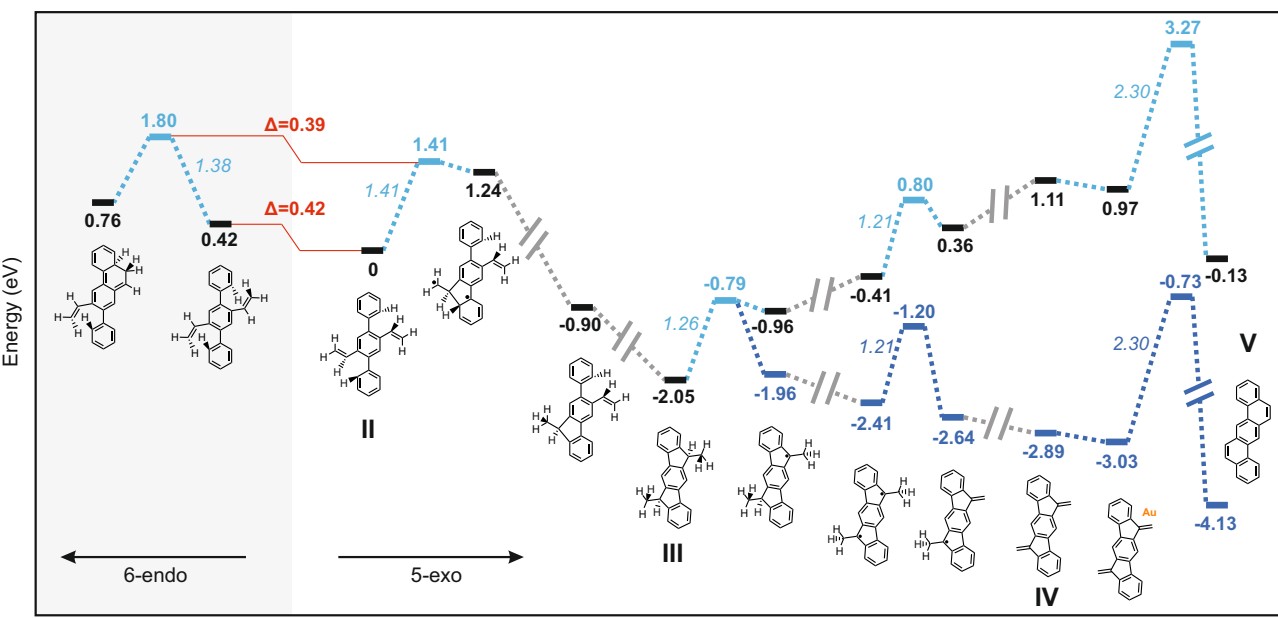

**Fig. 6 | Energy diagram of competing cyclization processes and the observed reaction cascade.** The gray background distinguishes the pathway for six-membered ring formation, with the reaction coordinate increasing from right to left. The reaction steps for five-membered ring formation and subsequent reactions have the reaction coordinate increasing from left to right and are on white background. The black bars and accompanying energy values depict the total energy of DFT-optimized geometries for various structures on Au(111), with corresponding chemical structures labeled. For dehydrogenation steps, detached hydrogen atoms are adsorbed on the gold substrate. Cyan bars and energy values denote transition states (bold cyan) and activation energies (italic cyan) for each reaction step, determined using the climbing image nudged elastic band (CI-NEB) method[54]. Gray interrupted lines represent reaction steps not modeled. Structures corresponding to experimentally observed species **2–5** are labeled **II-V**. Absolute energies after hydrogen removal are adjusted by 1 eV per split-off hydrogen to incorporate the entropy contribution of $H_2$ in vacuum[53], resulting in the dark blue energy landscape. Further details are provided in the "Methods" section.

imaging of the products obtained at each reaction stage unveiled a clear conversion sequence leading to the final polybenzo[*k*]tetraphene **5**. Vinyl groups first cyclized with the *ortho* phenylene rings at 155 °C, thus forming five-membered rings. This process produced exclusively the 5-*exo-trig* intermediate **3**, with reference to the classification in Baldwin's rules. After annealing to 215 °C, dehydrogenation featured dimethylene-dihydroindenofluorene structures (**4**), which were stable up to 380 °C. At this temperature, their conversion into benzo[*k*]tetraphene structures (**5**) was initiated, which completed after annealing to 405 °C. Apart from intermediate **3**, all the observed structures are stable over wide temperature ranges, allowing to obtain clean conversion from one architecture to the next with very high yield by annealing the sample to the corresponding temperature. Significantly, this study has provided a clear demonstration of the on-surface transformation from a pentagon to a hexagon, via a mechanism that involves gold adatoms. Moreover, the insights gained regarding the reactivity of surface-confined vinyl groups expand the repertoire of on-surface synthesis techniques, offering a valuable avenue for the creation of intricate and precise nanostructures in future research endeavors.

## Methods

### General methods for precursor synthesis
All reactions working with air- or moisture-sensitive compounds were carried out under argon atmosphere using standard Schlenk line techniques. Unless otherwise noted, all starting materials were purchased from commercial sources and used without further purification. All other reagents were used as received. Preparative column chromatography was performed on silica gel from Merck with a grain size of 0.063–0.200 mm (silica gel). Nuclear Magnetic Resonance (NMR) spectra were recorded in $CDCl_3$ or $CD_2Cl_2$ on AVANCE 300 MHz Bruker spectrometers. High-resolution mass spectra (HRMS) were performed on a SYNAPT G2 Si high-resolution time-of-flight mass spectrometer (Waters Corp., Manchester, UK) by matrix-assisted laser desorption/ionization (MALDI) using 7,7,8,8-tetra-cyanoquinodimethane (TCNQ) as matrix.

### STM/STS and nc-AFM experiments
The on-surface synthesis experiments were performed under ultrahigh vacuum (UHV) conditions with base pressure below $2 \times 10^{-10}$ mbar. Au(111) substrates (MaTeck GmbH) were cleaned by repeated cycles of $Ar^+$ sputtering (1 keV) and annealing (460 °C). The precursor molecules were thermally evaporated onto the clean Au(111) surface from quartz crucible heated at 120 °C with a deposition rate of ~0.5 Å·min$^{-1}$. STM images were acquired with a low-temperature scanning tunneling microscope (Scienta Omicron) operated at 4.7 K in constant-current mode using an etched tungsten tip. Bias voltages are given with respect to the sample. Constant-height dI/dV spectra were obtained with a lock-in amplifier (f = 610 Hz). nc-AFM measurements were performed at 4.7 K with a tungsten tip placed on a QPlus tuning fork sensor[44]. The tip was functionalized with a single CO molecule at the tip apex picked up from the previously CO-dosed surface[45]. The sensor was driven at its resonance frequency (22260 Hz) with a constant amplitude of 70 pm. The frequency shift from resonance of the tuning fork was recorded in constant-height mode using Omicron Matrix electronics and HF2Li PLL by Zurich Instruments. The Δz is positive (negative) when the tip-surface distance is increased (decreased) with respect to the STM set point at which the feedback loop is open.

### Computational details
All calculations were performed with AiiDAlab[46] apps based on the DFT code CP2K[47]. The surface/adsorbate system was modeled within the repeated slab scheme, with a simulation cell containing up to 1500 atoms, of which 4 atomic layers of Au along the [111] direction and a layer of hydrogen atoms to passivate one side of the slab in order to suppress one of the two Au(111) surface states. 40 Å of vacuum was included in the simulation cell to decouple the system from its periodic

replicas in the direction perpendicular to the surface. The size of the cell was $41.3 \times 51.1$ Å$^2$ corresponding to 140 Au(111) surface unit cells. The electronic states were expanded with a TZV2P Gaussian basis set[48] for C and H species and a DZVP basis set for Au species. A cutoff of 600 Ry was used for the plane waves basis set. Norm-conserving Goedecker-Teter-Hutter pseudopotentials[49] were used to represent the frozen core electrons of the atoms. We used the PBE parameterization for the generalized gradient approximation of the exchange-correlation functional[50]. To account for van der Waals interactions, we used the D3 scheme proposed by Grimme[51]. To obtain the equilibrium geometries, we kept the atomic positions of the bottom two layers of the slab fixed to the ideal bulk positions, and all other atoms were relaxed until forces were lower than 0.005 eV/Å. The Probe Particle model[52] was used to simulate nc-AFM images. In the energy landscape (Fig. 6) we computed the energy of all states without taking into account each of the entropic terms. We only included the entropic contribution after the dehydrogenation steps, following the assumption that the main contribution is the configurational entropy of the hydrogen atoms desorbing into vacuum as H$_2$, by far dominant with respect to other (e.g., vibrational) entropic terms[53]. The calculated energies and barriers are for monomeric units adsorbed on Au(111), emphasizing our focus on providing a qualitative overview of the reaction mechanism. This choice to compute reaction steps at the monomer level rather than longer chains is intentional, aiming to capture intramolecular processes unaffected by the polymer's long-range rigidity or flexibility. Activation barriers were estimated using the climbing image nudged elastic band (CI-NEB) method[54], with the experimentally visualized geometries serving as guides throughout the entire reaction path. Also in this case, all atoms in the adsorbates and first two layers of the substrate were allowed to relax with a convergence threshold on forces of 0.005 eV/Å. A detailed computational approach is outlined in Section 3 of the Supplementary Information. In all the reaction steps we could find a reasonable mechanism without the necessity to consider Au adatoms – which does not imply that they are not present/involved – except for the last transformation from **IV** to **V**. Here, although we only presented the initial stage of one of the plausible mechanisms, we were forced to include a gold adatom to reduce the energy barrier to a meaningful value. We highlight that a gold adatom was also included in each of the previously computed geometries, but was placed far from the molecular units. This allows a direct comparison of the energies in all steps. All calculations were conducted with molecular units adsorbed on Au(111).

## Data availability

The data reported in this study have been deposited in the Materials Cloud archive under accession code doi:10.24435/materialscloud:6f-kw[55]. All other data are available from the corresponding author upon request.

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

## Acknowledgements
This work was supported by the Swiss National Science Foundation under Grant No. 200020_182015 (R.F.) and 200020_212875 (R.F.), the NCCR MARVEL funded by the Swiss National Science Foundation (205602, C.A.P.), the European Union's Horizon 2020 research and innovation program under grant agreement number 881603 (Graphene Flagship Core 3, R.F.), the Max Planck Society, and the Alexander von Humboldt Foundation. Computational support from the Swiss Super-computing Center (CSCS) under project ID s1141 (C.A.P.) is gratefully acknowledged. We acknowledge PRACE for awarding access to the Fenix Infrastructure resources at CSCS, which are partially funded from the European Union's Horizon 2020 research and innovation program through the ICEI project under the grant agreement No. 800858 (C.A.P.). We are thankful to Lukas Rotach (Empa) for his excellent technical support during the experiments. S.A. and A.N. appreciate the help and support provided by the Scientific Computing and Data Analysis Section of Research Support Division at Okinawa Institute of Science and Technology Graduate University.

## Author contributions
M.D.G., P.R., K.M., A.N., and R.F. conceived the project. Z.Q. synthesized and characterized the precursor molecule. M.D.G. performed the STM/nc-AFM experiments. C.A.P. and S.A. performed the DFT calculations. All authors contributed to discussing the results and writing the manuscript.

## Competing interests
The authors declare no competing interests.
