## [Peer Review File · Nature Communications]

On-Surface Cyclization of Vinyl Groups on Poly-para-phenylene Involving An Unusual Pentagon to Hexagon TransformationReviewers' Comments:

Reviewer #1:

Remarks to the Author:

M. Di Giovannantonio and coworkers report an interesting new cyclization reaction for on-surface synthesis. They found that vinyl groups on poly-para-phenylene cyclize with the phenylene rings, forming a dimethyl-dihydroindenofluorene unit, which further dehydrogenates to a dimethylene-dihydroindenofluorene structure. The latter rearranges to form a polybenzo[k]tetraphene motif, involving a novel pentagon-to-hexagon transformation. The reaction steps occur in specific temperature windows. The reaction products were studied by STM/STS and insight into the reaction mechanism was obtained by DFT calculations. The experimental results and data analysis are of excellent quality and, together with the novelty of the results, justify publication in Nature Communication. There is, however, some room for improvement regarding the discussion of the computational part:

1) It is mentioned that the "absolute energies after hydrogen removal (reaction steps at ...) have been lowered by 1 eV per split-off hydrogen, to account for the entropy contribution of H₂ in vacuum." – This raises several questions: First, what is the meaning of the resulting energies in terms of thermodynamics? Due to the added entropy term, it is not the internal energy or the enthalpy, but also not the (Gibbs) free energy, because this would require that additional entropy contributions of the condensed phase (mainly vibrational entropy) are taken into account. Second, were entropy corrections also included in the transition state energies and if yes, how? If no entropy contributions were considered for the transition states, how does this affect the activation energies?

2) The barrier of 2.2 eV for the final rearrangement step is not consistent with the experimental temperature. Such a lack of agreement is usually cause to reject the proposed mechanism, but perhaps the authors can discuss reasons why it is still justified to keep this mechanism in this case (e.g., the semiquantitative level of the calculations).

3) The citations to the statement "recent years have witnessed the development of unique on-surface processes that are not easily accessible through traditional solution chemistry" seem somewhat arbitrarily chosen.

In summary, the study under review represents a considerable advance in the field of on-surface synthesis, is very innovative, and has excellent technical quality. Publication in Nature Communications is recommended after the authors have addressed the above comments.

Reviewer #2:

Remarks to the Author:

The use of scanning probe methods for on-surface reactions is a powerful tool to unravel mechanisms and discover new chemical reactions that lead to interesting conjugated polymers, such as graphene-like nanoribbons. The authors explored reactivity of vinyl group with phenyl ring on Au surface using temperature to activate the reactions. The results are unexpected, and cyclisation took place via a different mechanism that would normally be anticipated. By using a step-wise annealing protocol, the authors elucidated the exact mechanism of the cyclisation, by trapping intermediates, and showing an unexpected five to six-membered ring transformation. The experimental result and proposed mechanism are supported by theoretical modelling, which reveals a catalytic role of Au atoms in the process of C-C bond dissociation required for the rearrangement.

On a technical level, the paper is well executed. But I would like to clarify the following points:

1) what is the nature of branching points, sharp kinks, double chains in the polymer (Figure 1d)? These features are very abundant, but there is no discussion. The latter seems to be related to cross-

linking the chains through methyl groups (SI Fig 5b). This is an important part of the material characterisation;

2) are the calculated energies and barriers in Fig 6 for the polymer or monomer? The structural diagrams in this figure imply monomeric molecules. If this is the case, this point should be discussed carefully.

3) this is misleading for the readers to see temperature printed on the micrograph - the images are acquired at cryogenic temperature, and the temperatures stated were used for annealing prior to imaging. This should be clearly stated in the figures / captions.

Reviewer #3:

Remarks to the Author:

The manuscript submitted to Nature Communications by Prof. Fasel and co-workers illustrates the "On-Surface Cyclization of Vinyl Groups on Poly-para-phenylene Involving An Unusual Pentagon to Hexagon Transformation".

The manuscript reports an on-surface reaction illustrating the role of vinyl in the formation of 5-membered rings followed by conversion to 6-membered rings, which could be of interest to the surface science and nanostructured materials communities. There are some unclarified points or unsupported claims which would leave room for the manuscript to be further improved.

1. Is there any reaction of similar nature in solution chemistry? If yes, what are the reaction conditions, yield, and reaction mechanism, if known. I would suggest discussing the reported on-surface reaction in the manuscript in the context of comparing with solution chemistry, if available, to compare the similarities and differences.

2. It is not clear whether the presented reaction in this study offers a more promising, efficient, selective, and competitive route in constructing ladder-type nanostructures, compared to other previously studied routes. Comparison with other reactions in terms of readiness, efficiency, selectivity, etc is recommended.

3. As stated in the manuscript, the non-planar structure of the molecule and the out-of-planar geometry caused by the steric repulsion of the hydrogen atoms, many of the nc-AFM images are not conclusive nor clearly illustrative of the proposed structures. I am not convinced that readers would be able to retrieve the claimed analysis from the nc-AFM images. I can see this issue with nc-AFM images in Figures 2,3, starting from the imaged single molecule, Fig. 2b, 2c, where there is one brighter feature in the nc-AFM image, which seems to point to one vinyl group. If the nc-AFM images are not strongly supporting the claims, I would suggest moving them to the SI. The same problem remains with the two-dot bright features which constitute discussion surrounding a significant part of the results. There is a proposed mechanism for the reaction, which is on the basis of best matching the simulated structures, suggested as the intermediate state, with the nc-AFM images. See another related comment on the simulation data.

4. The simulated path using the method described is another crucial concern. The manuscript states that NEB simulation or String method was not carried out due to being computationally expensive. The authors have "decided to approximate the reaction path by means of constrained geometry optimizations".

This is concerning as the whole reaction path, analysis of the nc-AFM image of the intermediate structure, and energy profile are not based on an acceptable simulation on minimum energy path. This is particularly critical because the experimental data, nc-AFM (see comment #3) are not solely nor strongly conclusive of the proposed intermediate state.

The accuracy of the employed method and the energy should be quantitatively compared and

validated with the experimental data to justify the validity of the simulated approach. This analogy would be only possible had the study provided experimental data on surface reaction dynamics. Otherwise, an acceptable minimum energy path, e.g. NEB simulation, is necessary to justify the proposed reaction path and mechanism.

In the related context, it is not clear what the role of gold ad-atom is in the mechanism as the inclusion of Au-ad atom has changed the reaction's energy profile. For example, why gold-ad atoms are not playing any role in the steps involving the conversion of structure 1 -> 2 -> 3 ->4 (Fig. 1b) but they are included in the steps involving the conversion of 4 -> 5 (Fig. 5 i).

5. Would the dissociated iodine atoms have any impact in the reaction mechanism, for example by capturing hydrogen atoms from vinyl group? The manuscript states that a mild annealing to 75 C has led to the formation of 1D polymer (in which all carbon-iodine bonds have been dissociated. Fig. 3a (SI) shows that a self-assembled structure has likely formed. How much energy is required to destroy the self-assembly vs dissociating C-I bonds?

Where are the iodine atoms located after departing from the molecules, remaining on the surface or desorbing from the surface? How much is the diffusion energy of iodine atoms on the Au(111) surface?

Have the authors carried out any experiment on depositing the molecule at temperatures below RT and possibly tracking down where iodine atoms are gone?

6. It is stated in the manuscript that roughly 30% of the vinyl groups transformed into bright dot-like features whose identification constitutes a significant part of the manuscript. What happens to the remaining 70% of the vinyl groups which do not take part in transforming to the bright dot-like features? Would the transformation be kinetically fast and thus cannot be recorded by STM or nc-AFM, or else, the lack of transformation would have led to defects in the final structure or a reduced overall efficiency or reaction yield?

Resubmission

“On-Surface Cyclization of Vinyl Groups on Poly-para-phenylene Involving An Unusual Pentagon to Hexagon Transformation”

Point-by-point replies

Reviewer #1

M. Di Giovannantonio and coworkers report an interesting new cyclization reaction for on-surface synthesis. They found that vinyl groups on poly-para-phenylene cyclize with the phenylene rings, forming a dimethyl-dihydroindenofluorene unit, which further dehydrogenates to a dimethylene-dihydroindenofluorene structure. The latter rearranges to form a polybenzo[k]tetraphene motif, involving a novel pentagon-to-hexagon transformation. The reaction steps occur in specific temperature windows. The reaction products were studied by STM/STS and insight into the reaction mechanism was obtained by DFT calculations. The experimental results and data analysis are of excellent quality and, together with the novelty of the results, justify publication in Nature Communication. There is, however, some room for improvement regarding the discussion of the computational part:

Our reply: We are grateful to the reviewer for recognizing the value of our work and for the constructive remarks on the computational part. We have fully addressed them below, and made the corresponding descriptions in the manuscript clearer.

1) It is mentioned that the “absolute energies after hydrogen removal (reaction steps at ...) have been lowered by 1 eV per split-off hydrogen, to account for the entropy contribution of H₂ in vacuum.” – This raises several questions: First, what is the meaning of the resulting energies in terms of thermodynamics? Due to the added entropy term, it is not the internal energy or the enthalpy, but also not the (Gibbs) free energy, because this would require that additional entropy contributions of the condensed phase (mainly vibrational entropy) are taken into account. Second, were entropy corrections also included in the transition state energies and if yes, how? If no entropy contributions were considered for the transition states, how does this affect the activation energies?

Our reply: The reviewer is correct in pointing out that the description of our reported energy landscape needs further elucidation. Computing the Gibbs free energy would require to calculate the entropic terms (vibrational and configurational) at every step, which is only possible at the high cost of performing, for example, molecular dynamics simulations. This is out of the scope of our work, as we aim at offering a qualitative overview of the reaction mechanism. In the field of on-surface chemical reactions involving dehydrogenation steps, there exists an established approximation that deals with entropic terms and that has been widely accepted and adopted by the community. It was introduced by J. Björk in 2016 (J. Phys. Chem. C, 2016, 120, 21716–21721, cited as Ref. 36 in our manuscript) and is based on the assumption that the main entropic effect is represented by the configurational entropy of the hydrogen atoms desorbing into vacuum as H₂. In the ultrahigh vacuum conditions at which the experiments are performed, this energy gain amounts to about 1 eV for each split-off hydrogen atom.

As many reactions are characterized by several dehydrogenation steps (in our case 4 each polymer unit), this contribution sums up to a significant free-energy gain, and it becomes responsible of the thermodynamic behavior of the entire chemical reaction. As mentioned by Björk in his paper, "for studying the

thermodynamic aspects of this particular reaction, it appears sufficient to add the entropy of the hydrogen gas to the electronic enthalpy. This is a quite appealing approach for many problems, as the entropy of the hydrogen gas is added simply as an empirical correction, which does not add to the computational effort, while computing vibrational enthalpies and entropies requires quite a significant additional effort, even at the level of the harmonic approximation".

In our energy landscape, we have therefore taken into account the configurational entropy due to H₂ in vacuum only in the dehydrogenation steps, where such term is by far dominant as compared to other entropic terms (e.g. vibrational) present in these and all other reaction steps. The entropy was not taken into account in other steps (neither in the transition states, to answer the reviewer).

Action: We have added a new description in the Methods section of our manuscript to clarify all aspects regarding the entropy contribution to the energy landscape in the calculations that produced our Figure 6.

2) The barrier of 2.2 eV for the final rearrangement step is not consistent with the experimental temperature. Such a lack of agreement is usually cause to reject the proposed mechanism, but perhaps the authors can discuss reasons why it is still justified to keep this mechanism in this case (e.g., the semi-quantitative level of the calculations).

Our reply: We thank the reviewer to give us the chance to better clarify the scope of the energy landscape that we report in our manuscript. First, we would like to point out that energy barriers larger than 2 eV have also been reported in literature when computing the energy profiles of similar reactions (see, e.g., Chem. Sci., 2021, 12, 15637–15644, and J. Am. Chem. Soc. 2022, 144, 32, 14798–14808). Moreover, using the Arrhenius equation to estimate the rate constant of a chemical reaction, one finds $k=3.4E^{-4} \text{ s}^{-1}$ for reaction from **IV** to **V** (experimentally observed at 400 °C, with a computed barrier of 2.2 eV). Considering the experimental annealing time of >20 minutes, this gives about 0.4 events. Although being a low value, it still appears reasonable given such a simplified calculation.

In any case, we would like to stress that our description of the energy landscape occurs with two different regimes for the reaction steps **II**-to-**IV** and **IV**-to-**V**. In the first part of the energy profile (**II**-to-**IV**) we carefully describe the reaction mechanism including most of the transformation steps that lead to the formation of compound **IV**. On the other hand, in the case of the last step (**IV**-to-**V**) we do not claim a mechanism, as this is too complicated. Our aim is to offer a hint of how this reaction step could possibly proceed through an oversimplified calculation that only deals with an initial ring opening process. The full description of such complex ring rearrangement is out of the scope of our present work, and we leave it for a dedicated computational study.

Action: We have added a description to stress the scope of our energy landscape description in the revised manuscript, highlighting that the last step from **IV** to **V** is only reported as a speculation of its initial stage.

3) The citations to the statement “recent years have witnessed the development of unique on-surface processes that are not easily accessible through traditional solution chemistry” seem somewhat arbitrarily chosen.

Our reply: The citations corresponding to the reported statement include works reporting a reaction that was never observed in traditional solution chemistry (or proceeded in a significantly different manner). In the majority of the cases, on-surface synthesis studies have exploited knowledge already established in wet chemistry approaches. Hence, such reports are still few. We agree with the reviewer that the selected citations were not properly reflecting our statement.

Action: We have rephrased our statement to make it more correct, and updated the corresponding citations, also including a recently discovered [3+3] cycloaromatization between isopropyl groups.

In summary, the study under review represents a considerable advance in the field of on-surface synthesis, is very innovative, and has excellent technical quality. Publication in Nature Communications is recommended after the authors have addressed the above comments.

Our reply: We are extremely grateful to the reviewer for the acknowledgment of the innovative and high quality character of our work, and for recommending it for publication.

Reviewer #2 (Remarks to the Author):

The use of scanning probe methods for on-surface reactions is a powerful tool to unravel mechanisms and discover new chemical reactions that lead to interesting conjugated polymers, such as graphene-like nanoribbons. The authors explored reactivity of vinyl group with phenyl ring on Au surface using temperature to activate the reactions. The results are unexpected, and cyclisation took place via a different mechanism that would normally be anticipated. By using a step-wise annealing protocol, the authors elucidated the exact mechanism of the cyclisation, by trapping intermediates, and showing an unexpected five to six-membered ring transformation. The experimental result and proposed mechanism are supported by theoretical modelling, which reveals a catalytic role of Au atoms in the process of C-C bond dissociation required for the rearrangement.

On a technical level, the paper is well executed. But I would like to clarify the following points:

1) what is the nature of branching points, sharp kinks, double chains in the polymer (Figure 1d)? These features are very abundant, but there is no discussion. The latter seems to be related to cross-linking the chains through methyl groups (SI Fig 5b). This is an important part of the material characterisation;

Our reply: We thank the reviewer for highlighting the quality of our work, and for the comments that give us the opportunity to improve some of the descriptions. Referring to the first STM image in Figure 1d (acquired after annealing the sample to 75 °C), the reviewer is right in noting that molecular interconnections different from the targeted linear ones are present. They seem abundant due to the overall effect of the image to the eye, but in fact, they represent only a minority of the total coupling moieties. The desired dehalogenative aryl-aryl coupling leading to linear polymers represents the majority of interconnections between molecular units, which involves almost 90% of the active sites (two per molecular precursor). T-shaped connections are also observed (7% of the active sites), and we attribute them to the unwanted Heck reaction between a carbon radical (after deiodination) and a vinyl group. Indeed, we used iodine as leaving group installed at the active sites of our precursor molecules to suppress the Heck reaction, which would have been much more effective with bromine. In fact, the debromination temperature of aryl halides on Au(111) is higher than the deiodination one (about 170 °C vs 70 °C, respectively, see e.g. ACS Nano 2018, 12, 74–81), promoting an enhanced activation of the vinyl groups towards the Heck reaction. By employing iodine, we successfully managed to limit this side reaction to 7%. There are also very few sites with still iodine linked to the molecule (2%), due to the moderate annealing temperature, and some triple connections (2%) that we did not investigate further. Regarding the "double chains" pointed out by the reviewer (e.g. top of Figure 1d left), they are not due to a covalent

coupling between two polymer strands (as instead is the case of Supplementary Fig. 5), but to a weak interaction between them. This interaction is sometimes mediated by iodine atoms placed in between the chains, via halogen-hydrogen stabilization. This type of assembly is observed after cooling the sample down to 4.7 K for imaging, but most likely, it does not survive room temperature, while covalent interchain coupling is only observed after annealing at higher temperature (e.g. Figure S5, obtained after annealing the sample to 380 °C).

The statistical analysis that we discussed for the first STM image in Figure 1d is similarly reflected in the three remaining samples (annealed at higher temperature). The T-shaped connections occur in less than 10% of the active sites, while the iodinated sites and triple connections are absent, leaving more than 90% of the targeted intermolecular connections. The only exception is the appearance of some covalent couplings between adjacent chains in the last sample, due to non-selective CH activation occurring above 300 °C (see, e.g. Supplementary Fig. 5).

Action: We have added a description of the observed side products in the text of our revised manuscript, and highlighted them in the new Fig. 1d.

2) are the calculated energies and barriers in Fig 6 for the polymer or monomer? The structural diagrams in this figure imply monomeric molecules. If this is the case, this point should be discuss carefully.

Our reply: The calculated energies and barriers reported in Figure 6 are obtained for monomer units. Our aim is to provide a qualitative overview of the reaction mechanism. Computing reaction steps that occur at monomer units instead of longer chains is still representative as we focus on intramolecular processes that are not expected to be significantly affected by the rigidity/flexibility of the polymer on a long range. Moreover, this approach allows to study many reaction steps within a reasonable time frame, whereas it would call for a dedicated computational study if one used longer polymeric chains to compute every step.

Action: We are grateful to the reviewer for highlighting this unclear point, which is important to clarify. In the Methods section of the revised version of our manuscript, we have added an accurate description of the computed entities and justification of the approach.

3) this is misleading for the readers to see temperature printed on the micrograph - the images are acquired at cryogenic temperature, and the temperatures stated were used for annealing prior to imaging. This should be clearly stated in the figures / captions.

Our reply: We thank the reviewer for this comment. Reporting temperatures on the images is commonly done to help the reader to quickly recognize the sample characteristics and how it was processed, which often implies thermal heating steps.

Action: To avoid confusion, we have now added precise statements about the processing and image acquisition temperature in each of the captions of our revised manuscript.

Reviewer #3 (Remarks to the Author):

The manuscript submitted to Nature Communications by Prof. Fasel and co-workers illustrates the “On-Surface Cyclization of Vinyl Groups on Poly-para-phenylene Involving An Unusual Pentagon to Hexagon Transformation”.

The manuscript reports an on-surface reaction illustrating the role of vinyl in the formation of 5-membered rings followed by conversion to 6-membered rings, which could be of interest to the surface science and nanostructured materials communities. There are some unclarified points or unsupported claims which would leave room for the manuscript to be further improved.

1. Is there any reaction of similar nature in solution chemistry? If yes, what are the reaction conditions, yield, and reaction mechanism, if known. I would suggest discussing the reported on-surface reaction in the manuscript in the context of comparing with solution chemistry, if available, to compare the similarities and differences.

Our reply: We are grateful to the reviewer for this comment, which allows us to put our results in a better context. Indeed, there exist similar cyclization reactions in solution chemistry, as discussed below. Intramolecular arylation of alkynes can be catalyzed by transition metals or Lewis acids (J. Am. Chem. Soc. 2008, 130, 5636–5637, ref. 31 cited in our original manuscript). The outcome of such intramolecular cyclization reactions can be divided into *exo*- and *endo*-types, whose kinetically favored products have been discussed as Baldwin’s rules. For example, depending on the reaction conditions, intramolecular alkyne arylation of 1-phenyl-2-ethynylbenzene (**S4**) can lead to either phenanthrene (**S6**) via a 6-*endo-dig* ring closure or 9-methylene-9*H*-fluorene (**S9**) through a 5-*exo-dig* cyclization in high yields (Scheme R1a). It is worth mentioning that the hypervalent iodine (HVI)-mediated oxidative cyclization of 2-vinylbiphenyl (**S10**) is reported by Murphy group to yield the phenanthrene (**S6**) (Chem. Eur. J. 2018, 24, 17002–17005), while the Pd-catalyzed cyclization under aerobic oxidation conditions can produce the 9-methylene-9*H*-fluorene (**S9**) (Org. Lett. 2023, 25, 800–804) (Scheme R1b). The reaction mechanism of the latter case can be similar to the first step in our work, where the gold adatom could possibly mediate the cyclization reaction. Notably, our system experiences a pentagon-to-hexagon transformation, which has never been described in solution chemistry. The 2-vinylbiphenyl (**S10**) can also undergo photochemical cyclization in both the absence and the presence of oxygen to produce 9,10-dihydrophenanthrene (**S15**) in quantitative yield (J. Org. Chem., 1973, 38, 3801–3803). Morgan *et al* proposed the unstable 8a,9-dihydrophenanthrene intermediate (**S14**), which then undergoes a thermal sigmatropic hydrogen shift to yield the observed product (Scheme R1c). Remarkably, phenanthrene is not observed during this photocyclization, and thus the mechanism and product are very different from our system.

Action: Based on the discussions reported above, the comparison between our on-surface reaction and solution chemistry analogues has been extended in the Introduction of the revised manuscript. Moreover, we have added the Scheme R1 as a new scheme in the supplementary material (new Supplementary Scheme 2).

Scheme R1. Similar cyclization reactions in solution chemistry.

2. It is not clear whether the presented reaction in this study offers a more promising, efficient, selective, and competitive route in constructing ladder-type nanostructures, compared to other previously studied routes. Comparison with other reactions in terms of readiness, efficiency, selectivity, etc is recommended.

Our reply: As discussed in the Introduction, the extension of an armchair edge is an important approach to construct graphene nanoribbons (GNRs) with novel architectures and exciting properties (e.g. narrow bandgaps or topologically protected states), yet the available toolkit leading to clean conversion is rather limited. For example, the on-surface methyl–methyl coupling only worked in the case of small molecules to produce circumcoronene (Science Advances, 2021, 7, eabf0269), but failed to yield the complete reaction towards the targeted edge-extended 9-AGNR (Organic Materials, 2021, 3, 128–133). Given the challenge in achieving such extension of the armchair edge, our innovative approach offers a valuable tool not only to improve the ladderization, but also for fabricating novel, low-dimensional nanostructures with high precision and yield.

Action: In the revised manuscript, we have enriched the discussion in the Introduction, and compared our approach to others, such as the methyl–methyl coupling.

3. As stated in the manuscript, the non-planar structure of the molecule and the out-of-planar geometry caused by the steric repulsion of the hydrogen atoms, many of the nc-AFM images are not conclusive nor clearly illustrative of the proposed structures. I am not convinced that readers would be able to retrieve the claimed analysis from the nc-AFM images. I can see this issue with nc-AFM images in Figures 2,3, starting from the imaged single molecule, Fig. 2b, 2c, where there is one brighter feature in the nc-AFM image, which seems to point to one vinyl group. If the nc-AFM images are not strongly supporting the claims, I would suggest moving them to the SI. The same problem remains with the two-dot bright features which constitute discussion surrounding a significant part of the results. There is a proposed mechanism for the reaction, which is on the basis of best matching the simulated structures, suggested as the intermediate state, with the nc-AFM images.

See another related comment on the simulation data.

Our reply: In our work, nc-AFM images represent the key experimental tool to identify the chemical structure of the reported nanomaterials. Being a (mainly) constant-height technique, where the distance between tip and sample is constant during the image acquisition, planar nanostructures are more easily visualized. Nevertheless, also in the tricky cases of nonplanar compounds, imaging at varying tip-sample distances, imaging at constant current, and the comparison with simulated nc-AFM images are valid and solid means to achieve an unambiguous structural identification. In this regards, simulated nc-AFM images do not rely on assumptions or approximations, but are based on standard DFT relaxations and have proven to be highly reliable in supporting the interpretation of experimental nc-AFM images.

In our study, we characterized five compounds by means of STM and nc-AFM imaging, and the latter provided clear experimental evidence of the structure of four of them (**1**, **2**, **4**, and **5**). Compound **3**, due to the severe height difference between the "dot-like" features and the rest of the chain, required an extensive investigation also from the theory point of view, by comparing its nc-AFM image with several simulated structures (discussed in our Supplementary Fig. 4). Such in-depth analysis led to a clear assignment of the structure of **3**, which relies not only on the best match between experimental and simulated nc-AFM images, but also on the lower energy of the assigned geometry with respect to others, as computed by standard DFT.

We agree with the reviewer that the identification of compounds **1** and **2** is certainly less straightforward than that of compounds **4** and **5**, due to the out of plane structures. However, these difficulties are common, and also a simple compound such as poly(*para*-phenylene) presents similar twists that could make its structural identification less obvious (see, e.g., Nat. Synth. 2022, 1, 289–296). However, the nc-AFM community has so far profited from many examples and experts can identify even very complex and highly nonplanar compounds (see, e.g., Angew. Chem. Int. Ed. 2018, 57, 3888–3908; Astrophys. J. Lett. 2021, 908, L13; Angew. Chem. Int. Ed. 2023, 62, e202218211). To help the readers, we have anyway added two new nc-AFM simulations in Fig. 2, and reshaped the entire figure layout.

Action: We have now added two new nc-AFM simulations in Figure 2, which better elucidate the chemical structures of compounds **1** and **2**. This will guide the readers to better retrieve the assigned geometries also in these less straightforward cases.

4. The simulated path using the method described is another crucial concern. The manuscript states that NEB simulation or String method was not carried out due to being computationally expensive.

The authors have “decided to approximate the reaction path by means of constrained geometry optimizations”.

This is concerning as the whole reaction path, analysis of the nc-afm image of the intermediate structure, and energy profile are not based on an acceptable simulation on minimum energy path. This is particularly critical because the experimental data, nc-afm (see comment #3) are not solely nor strongly conclusive of the proposed intermediate state.

Our reply: Here, we would like to clarify our approach, to avoid confusion. The reviewer is incorrect when stating "*the whole reaction path, analysis of the nc-afm image of the intermediate structure, and energy profile are not based on an acceptable simulation on minimum energy path*". Specifically, two points are wrong. First, the analysis of the nc-AFM images of the intermediate structures is not based on the simulation of minimum energy paths. As explained in our reply to the previous point, nc-AFM simulations are obtained from DFT-based geometry relaxations, which is a standard and well-established method to complement (and interpret) experimental data. Second, the statement that our reaction path and energy profile are not acceptable because not based on the climbing image nudged elastic band (CI-NEB) method is questionable. We stress that, to derive the energy landscape reported in our Figure 6, we have performed a series of constrained geometry optimizations based on DFT. This approach is certainly less accurate than the NEB method in the identification of the exact energy of the transition state, but also less computationally demanding. Our main objective is to provide a qualitative overview of a possible reaction mechanism, and not to pinpoint accurate energy values of states and barriers, which is out of the scope of the present study and better suited for a dedicated computational study. Moreover, the NEB method is based on knowing the initial and final state of each step of a reaction and is strongly biased by the initial guess of the reaction trajectory that must be provided as starting point for the simulation. Failing in the identification of the correct trajectory leads to a wrongly determined reaction path. In this regard, we highlight that the proposed reaction mechanism takes the experimentally observed stable intermediates and products as solid cornerstones, and we only present plausible reaction mechanisms from one to the other. The path that we build from one to the other with a series of constrained optimizations is highly unbiased since it is obtained imposing only one constraint on the initial state and verifying that the smooth variation of the constraint would lead to the desired final state.

Action: In the Computational details section of the Methods, we have added a new description of the approach that we used to achieve the energy landscape reported in Figure 6.

The accuracy of the employed method and the energy should be quantitatively compared and validated with the experimental data to justify the validity of the simulated approach.

This analogy would be only possible had the study provided experimental data on surface reaction dynamics. Otherwise, an acceptable minimum energy path, e.g. NEB simulation, is necessary to justify the proposed reaction path and mechanism.

Our reply: We take the chance to further clarify our approach by answering to this additional comment on the theory part of our manuscript. First, we would like to distinguish between the reaction path and mechanism. The reaction path necessarily goes through the stable intermediate states and products, which we observe experimentally and thus identify without the need to refer to theory/calculations. On the other hand, we described the reaction mechanism by theoretically proposing plausible reaction cascades in between the experimentally observed intermediates/products. Our aim is to furnish the readers with a qualitative overview of the most plausible mechanisms that can reproduce the experimentally observed results. Moreover, as we describe in our reply to the previous point, the method that we used to determine the energy landscape is as acceptable as the NEB method.

Action: These concepts have now been added and carefully described in our revised manuscript (Computational details section of the Methods).

In the related context, it is not clear what the role of gold ad-atom is in the mechanism as the inclusion of Au-ad atom has changed the reaction's energy profile.

For example, why gold-ad atoms are not playing any role in the steps involving the conversion of structure 1 -> 2 -> 3 ->4 (Fig. 1b) but they are included in the steps involving the conversion of 4 -> 5 (Fig. 5 i).

Our reply: In Figure 6 of our manuscript, we are not claiming to offer the one and only correct reaction mechanism, but we are proposing a plausible (and partial) one. There might be other mechanisms equally favorite, but the identification of the absolute minima and the description of the complete process (which involves many steps) are out of the scope of the present study. Specifically, as in our reply to Reviewer 1, point 2, we highlight that, for the transformation from **IV** to **V**, we are not claiming any mechanism, but simply commenting on the possible initial stage of the ring opening. In the spirit of finding a plausible route, we were forced to take into account the interplay of a gold adatom in such ring rearrangement step to justify the C-C bond scission with a reasonable energy barrier. We speculate that the role of such Au adatom could be to enhance an electron transfer from the substrate to the molecule and promote the ring opening. In all the previous steps we could find a reasonable mechanism without the necessity to consider Au adatoms, which does not imply that they are not present/involved. In fact, we have tested the initial cyclization step with and without Au adatom, which produced a qualitatively similar result (Fig. S6). Finally we stress that, although not properly indicated in Figure 6, all the DFT-optimized geometries used to determine the energy landscape presented a Au adatom on the surface, far from the molecular species. Only for step **IV** to **V** (where we placed the orange label "Au") such Au adatom was placed in proximity of the bond that is broken. This ensures that the total energies are consistent throughout all the steps. The same was done for the simulations reported in Figure S6.

Action: We have now added a dedicated statement in the Computational details section of the Methods, clarifying the role of gold adatoms. Moreover, to avoid confusion, we have carefully described how they were used in the simulations of all steps and placed at different locations with respect to the molecular species according to their expected role.

5. Would the dissociated iodine atoms have any impact in the reaction mechanism, for example by capturing hydrogen atoms from vinyl group?

The manuscript states that a mild annealing to 75 C has led to the formation of 1D polymer (in which all carbon-iodine bonds have been dissociated.

Fig. 3a (SI) shows that a self-assembled structure has likely formed. How much energy is required to destroy the self-assembly vs dissociating C-I bonds?

Our reply: Although less frequently than bromine, iodine atoms are often used in on-surface synthetic studies as leaving groups in dehalogenative aryl-aryl coupling processes. After being detached from the molecular precursors, they chemisorb onto the metal surface as byproducts. Typically, they have no further impact in the on-surface processes, with the only exceptions of (i) promoting the self-assembly of organic species via halogen-hydrogen bonding, and (ii) capturing atomic hydrogen that is possibly produced by subsequent reaction steps, and desorbing as HX (Small 2009, 5, 592–597; J. Phys. Chem. C 2015, 119, 486–493; ACS Nano 2018, 12, 74–81; Chem. Sci. 2019, 10, 2998–3004). In our case, we hypothesize a similar behavior. In fact, we observe iodine in between adjacent polymer strand when imaging the sample after annealing to 75 °C (Figure 2d). Moreover, we notice a reduction in the number of iodine atoms stabilized at the elbow sites of the herringbone reconstruction of the Au(111) surface after annealing to 215 °C (compare the evolution of the dots observed in the STM images in Figure 2d). After annealing the sample to this temperature, the dehydrogenation step transforming **3** into **4** occurs, and the released atomic hydrogen could be captured by the available iodine atoms and desorb as HI. The self-assembly structure observed in Figure S3a is only observed after cooling the sample down to 4.7 K for imaging, but it is most likely absent at room temperature. We therefore speculate that the

energy required to destroy the self-assembly is lower than the one required to dissociate the C-I bonds (about 0.7 eV, according to J. Am. Chem. Soc. 2013, 135, 5768–5775).

Action: We have added a new description on the role of iodine during the investigated reaction in our revised manuscript.

Where are the iodine atoms located after departing from the molecules, remaining on the surface or desorbing from the surface? How much is the diffusion energy of iodine atoms on the Au(111) surface?

Our reply: As described in our reply to the previous point, iodine atoms initially chemisorb onto the Au(111) surface after detaching from the molecular precursors. Imaging of the samples at 4.7 K shows these iodine atoms as preferentially adsorbed at the elbow sites of the herringbone reconstruction of the Au(111) surface, or stabilized next/in between polymer chains. Their amount reduces after the annealing step at 215 °C as they likely desorb as HI after recombination with the atomic hydrogen released during the dehydrogenation step (from **3** to **4**). Their diffusion energy on Au(111) amounts to about 0.1 eV (J. Am. Chem. Soc. 2013, 135, 5768–5775).

Action: We have included this information in the newly added part regarding the role of iodine.

Have the authors carried out any experiment on depositing the molecule at temperatures below RT and possibly tracking down where iodine atoms are gone?

Our reply: The role and fate of iodine atoms in on-surface synthetic processes has been extensively investigated over the past decade, as described in our replies to the previous two points. We did not perform the experiment suggested by the reviewer. However, based on the acquired knowledge on this type of processes, we can state that during the annealing step that induces their detachment from the molecular precursors, the iodine atoms chemisorb on the Au(111) surface and have enough energy to diffuse around. They are then stabilized in preferred adsorption sites (e.g. at the elbow sites of the herringbone reconstruction of the Au(111) surface, or stabilized next/in between polymer chains) after cooling the sample down to 4.7 K for imaging. During the annealing steps at higher temperature, iodine atoms desorb initially as HI and then presumably as I₂ or AuI.

Action: We have included this information in the newly added part regarding the role of iodine.

6. It is stated in the manuscript that roughly 30% of the vinyl groups transformed into bright dot-like features whose identification constitutes a significant part of the manuscript. What happens to the remaining 70% of the vinyl groups which do not take part in transforming to the bright dot-like features? Would the transformation be kinetically fast and thus cannot be recorded by STM or nc-AFM, or else, the lack of transformation would have led to defects in the final structure or a reduced overall efficiency or reaction yield?

Our reply: We thank the reviewer for this comment, which gives us the opportunity to better clarify our statement. After annealing the sample to 155 °C, the system is trapped in a partially converted phase. Most of the polymer units still feature unreacted vinyl groups (70%), while 30% of these groups have transformed into the "dot-like" moieties. Annealing the sample to the same temperature for a longer time, or to a slightly higher temperature, would most likely lead to the observation of the "dot-like" features in all polymer units. However, our subsequent annealing step was performed at 215 °C, which already induced the conversion of all units into the planar compound **4**.

Action: In our revised manuscript, we have specified what happens to the remaining 70% of the vinyl groups, to avoid confusion.

Reviewers' Comments:

Reviewer #1:

Remarks to the Author:

The revisions have elevated an already impressive manuscript to an outstanding level. Nevertheless, there is one important aspect related to entropy corrections that remains unclear.

In my original comments, I raised the question "If no entropy contributions were considered for the transition states, how does this affect the activation energies?"

As far as I see, this question has not been addressed. It is only stated that the transition state energies have indeed not been corrected for entropy. However, if the energies of the stable states (i.e., potential minima, not transition states) have been lowered by 1 eV per removed hydrogen atom, but the transition states remain unchanged, would this not result in an unrealistic overestimation of the activation barriers?

Conversely, if the energies of the transition states are also lowered by the same amount, it should result in an underestimation of the barrier, depending on the extent of entropy already acquired in the transition state (and, thus, on the specific structure of the transition state).

Given the considerable magnitude of these entropy corrections, I think it is imperative that these aspects are addressed and the limitations of this approximation are thoroughly discussed.

The authors may also consider showing only the energy data (without entropy contributions) in the diagram and addressing the entropy contributions separately within the text.

Reviewer #2:

Remarks to the Author:

The authors have responded very well to all questions raised by the reviewers. The paper is much improved now and can be published in its current form.

Reviewer #3:

Remarks to the Author:

The current manuscript reports the study carried out by Prof. Fasel and co-workers, titled "On-Surface Cyclization of Vinyl Groups on Poly-para-phenylene Involving an Unusual Pentagon to Hexagon Transformation". The study is of interest to the surface science and nanostructured materials communities.

The experimental section is supported by high quality data taken by LT-STM and nc-AFM, displaying the role of vinyl in the formation of 5-membered rings followed by conversion to 6-membered rings.

I appreciate the modifications made by the co-authors to improve and clarify the data, presented. However, I still find some statements or conclusion made from the theoretical section remain to be unsupported.

1. I have one remained question for the experimental section.

Lines 117-118 (Page 4) of the manuscript state that intact precursor molecules were observed, either isolated or packed into clusters.

Have the co-authors recorded more images as to evaluate the statistics of whether 100% of the deposited molecules have been intact when adsorbed on the surface.

To clarify that the clusters consist of intact, and not dehalogenated molecules, I would suggest that co-authors analyze a few large-scale images and show a zoom-in image of clusters presented in the STM image of SI Fig. 3a. As well, please cite literature for the dissociation temperature of C-I on Au(111).

2. Barrier height/energy barrier is attributed to the energy required for the initial states to convert to transition state along the minimum energy pathway. Many plausible pathways can be calculated by varying different factors, for example bond length/distance, rotation angle, etc, but not all the structures with the highest energy for different plausible pathways are referred to as transition states, but only the one which is along the minimum energy path. The energy of the transition state is associated with one structure, namely the saddle point, along the minimum energy path. In the manuscript, the current theoretical data refer to several energies as 'reaction barrier' or 'energy barrier'. Co-authors justify that the presented path is not the minimum energy path, nor the presented mechanism is the only favourable mechanism, but 'a qualitative overview of the reaction mechanism', or 'a qualitative overview of a possible reaction mechanism, and not to pinpoint accurate energy values of states and barriers, which is out of the scope of the present study and better suited for a dedicated computational study.'

If NEB simulation or String method has been considered to be expensive within the resources available to the co-authors, I would suggest that the minimum energy path should be retrieved from calculating potential energy surfaces, at single energy points of molecular structure considering sufficient variables to be able to assign the saddle point, hence, retrieving the energy of the transition state without going through expensive simulation. Otherwise, the current calculations merely show random paths which the molecules might (or not) have taken.

Except the calculated structures which correspond to the experimental data – structures formed after the sequential annealing steps – the context of the theoretical data presented in this manuscript is imprecise (compared to the acceptable methods in literature) and misleading to the readers of Nature Communication journal.

3. Line 280-305 (Page 9) refer to the C-C bond breaking (SI Fig. 10) and that the energy barriers of 3 eV and 2.2 eV have been calculated in the absence and presence of gold ad-atom, respectively.

As it has been explained in the manuscript, the 2.2 energy is retrieved from the calculation where gold ad-atom is involved in the bond breakage. I assume the calculation of C-C bond breaking with 3 eV energy height has been carried out for the molecule adsorbed on the surface. I would suggest that for this calculation, one gold ad-atom should be similarly located on the surface but in further proximity to the molecule.

4. In the Page 9, it is written, 'Polymer 4 is very stable and experimentally observed over a wide temperature range'. Then a few lines later, it is stated that, 'The calculated energy barrier of 2.2 eV with adatoms is high enough to justify the stability of polymer 4 below 380 °C, which ensures an extremely high selectivity between two different cyclized products.'

Fig. 6 shows that structure IV has been stabilized by the presence of gold ad-atom (from -2.89 eV to -3.03 eV), which has been converted to the structure V with -4.13 eV. What are the two cyclized products referred here and the attribution to selectivity? For selectivity, there should be at least two competitive pathways to be compared with, hence, concluding one pathway leading to one product is more selective than the other one (starting from the same initial state). This is supported by having the minimum energy path for both, and comparing their barrier height.

For selectivity, do the co-authors mean a comparison between 5-exo and 6-endo?

In the Conclusion section, Page 10, it has been stated that structures 4 and 5 are the kinetically and

thermodynamically favoured products. In Fig. 6, I don't see any competitive path to the formation of structure 4. Thus, why structure 4 is concluded to be a kinetically favoured product, compared to what other structure?

Please clarify.

5. I also noticed that there are several 'energy heights' with 1.3 eV. I am curious if co-authors can justify it, as it may suggest a systematic repetition of a factor involved.

Resubmission

“On-Surface Cyclization of Vinyl Groups on Poly-para-phenylene Involving An Unusual Pentagon to Hexagon Transformation”

Point-by-point replies

Reviewer #1 (Remarks to the Author):

The revisions have elevated an already impressive manuscript to an outstanding level. Nevertheless, there is one important aspect related to entropy corrections that remains unclear.

In my original comments, I raised the question "If no entropy contributions were considered for the transition states, how does this affect the activation energies?"

As far as I see, this question has not been addressed. It is only stated that the transition state energies have indeed not been corrected for entropy. However, if the energies of the stable states (i.e., potential minima, not transition states) have been lowered by 1 eV per removed hydrogen atom, but the transition states remain unchanged, would this not result in an unrealistic overestimation of the activation barriers?

Conversely, if the energies of the transitions states are also lowered by the same amount, it should result in an underestimation of the barrier, depending on the extent of entropy already acquired in the transition state (and, thus, on the specific structure of the transition state).

Given the considerable magnitude of these entropy corrections, I think it is imperative that these aspects are addressed and the limitations of this approximation are thoroughly discussed.

The authors may also consider showing only the energy data (without entropy contributions) in the diagram and addressing the entropy contributions separately within the text.

Our reply: We appreciate the reviewer's assessment of our manuscript's quality and their valuable feedback on enhancing our discussion regarding entropy correction in transition states. In our energy landscape analysis, we have factored in the entropy correction exclusively for the potential minima following the dehydrogenation process – not for the transition states, where hydrogen remains present. As elucidated in our prior point-by-point response, this decision involved reducing the absolute energy of each final state (post each hydrogen loss) by 1 eV (as outlined in J. Björk, J. Phys. Chem. C, 2016, 120, 21716–21721). This choice primarily stems from the significant impact of hydrogen desorption into a vacuum on the entropy contribution, which becomes evident only upon hydrogen desorption from the surface. During our computations of the energy barriers in the dehydrogenation steps, each detachment of hydrogen from a carbon atom resulted in its adsorption on the gold substrate. Consequently, the energy barriers and transition states for these steps remain unaffected by the entropy gain from hydrogen desorption. This aspect was considered solely in the stable geometry calculated after a dehydrogenation step. We emphasize that, as detailed in our previous point-by-point response, while vibrational and configurational entropy corrections would impact all steps and transition states, their computation extends beyond the scope of our present study. Here, our focus was specifically on considering the entropy gain

due to hydrogen desorption post-detachment from the molecular units as the primary entropic factor influencing the thermodynamic behavior of the entire chemical reaction.

Action: To clarify our energy landscape and the energy corrections that were considered, we have now enhanced the graph in Figure 6 with a new one, following the reviewer's suggestion of showing the uncorrected energy values. Here, the uncorrected energies of each state are reported in black, and the computed energy barriers and transition states in cyan. The grey barriers and transition states indicate two steps that we did not compute but assumed to be the same as the previous ones, as they involve the same dehydrogenation process occurring at the opposite side of the molecular unit. In this new graph, we have added in dark blue the corrected energy profile. To clarify it, we hereby describe in detail some of the steps.

Starting from the state at -2.05 eV, we obtained the first dehydrogenation energy barrier of 1.3 eV, leading to a transition state at -0.75 eV. The next geometry that was optimized presented an energy of -0.96 eV with the detached H adsorbed on the gold substrate (far from the molecule). We have then lowered this energy by 1 eV (to -1.96 eV) to account for the desorption of the detached H into vacuum. This correction produces a lowering of the next transition state (from 0.34 eV to -0.66 eV) because the energy barrier is unaffected by such entropy correction and still amounts to 1.3 eV. In other words, we have set a new reference energy after the correction. The second dehydrogenation step leads to a geometry with uncorrected energy of -0.41 eV, which now includes two H atoms adsorbed on the gold substrate. Hence, we have lowered this energy by 2 eV (to -2.41 eV) to account for the desorption of two H atoms into the vacuum, and so on.

We think that showing both the uncorrected and corrected energy profiles is indeed a valid and clear way to describe the process and let the reader understand why the process proceeds towards the final, thermodynamically stable state (i.e. the all-hexagon phase). Moreover, the entropy correction shows that the energy barriers for the reverse process (from right to left of the energy diagram) increase, rendering it irreversible. This is consistent with the desorption of the detached hydrogen atoms into the vacuum, which cannot be reverted.

Reviewer #2 (Remarks to the Author):

The authors have responded very well to all questions raised by the reviewers. The paper is much improved now and can be published in its current form.

Our reply: We thank the reviewer for pointing out the significant improvement of our manuscript, and for accepting it for publication.

Reviewer #3 (Remarks to the Author):

The current manuscript reports the study carried out by Prof. Fasel and co-workers, titled "On-Surface Cyclization of Vinyl Groups on Poly-para-phenylene Involving an Unusual Pentagon to Hexagon Transformation". The study is of interest to the surface science and nanostructured materials communities.

The experimental section is supported by high quality data taken by LT-STM and nc-AFM, displaying the role of vinyl in the formation of 5-membered rings followed by conversion to 6-membered rings.

I appreciate the modifications made by the co-authors to improve and clarify the data, presented. However, I still find some statements or conclusion made from the theoretical section remain to be unsupported.

Our reply: We are grateful to the reviewer for the appreciation of the improvements made, and further worked to clarify the theoretical part, as outlined in the following.

1. I have one remained question for the experimental section.

Lines 117-118 (Page 4) of the manuscript state that intact precursor molecules were observed, either isolated or packed into clusters.

Have the co-authors recorded more images as to evaluate the statistics of whether 100% of the deposited molecules have been intact when adsorbed on the surface.

To clarify that the clusters consist of intact, and not dehalogenated molecules, I would suggest that co-authors analyze a few large-scale images and show a zoom-in image of clusters presented in the STM image of SI Fig. 3a. As well, please cite literature for the dissociation temperature of C-I on Au(111).

Our reply: We thank the reviewer for this comment. As we write in the manuscript, "A few molecules already formed dimers and trimers, indicating that deiodination was initiated already at RT [...]. However, the majority of the molecules were still intact". The attribution of intact species in the clusters (STM image in Supplementary Fig. 3a) is based on the morphology of the molecular units, which presents the same shape and size of the one imaged by high resolution STM and nc-AFM in Fig. 2a-f. Here, two iodine atoms are connected to the ends of the molecular building block. The experimentally measured distance of 2.3 ± 0.2 Å between I and C (from nc-AFM) matches well with the DFT calculated value of 2.14 Å for the molecule adsorbed on Au(111) and is clearly shorter than the C-I distance of 2.52 Å, found as transition state during deiodination of iodobenzene on Au(111) (*J. Am. Chem. Soc.* **2013**, *135*, 5768–5775). These observations confirm that the molecule is intact, and therefore also those in the clusters. The molecules in the clusters form two different self-assemblies (SA), as illustrated in Fig. R1. In SA1, the intact molecules are densely packed and very likely stabilized by halogen-hydrogen interactions between the iodine atoms and the vinyl groups. In SA2, the molecular packing is less dense, and some additional features (red circles) are visible in between the molecules – most likely iodine atoms detached from already reacted species.

According to the reviewer's suggestion, we extended the statistical analysis of intact molecules. From 140 molecules analyzed, we find that 20% of molecular sites are deiodinated, rationalizing the presence of iodine atoms chemisorbed on Au in between the molecular islands and the appearance of dimers and trimers, as discussed in the main text of our manuscript.

The C-I dissociation temperature on Au(111) depends on the molecular system and typically ranges from RT to 80 °C (*ACSNano* **2014**, *8*, 7880-7889; *ACSNano* **2018**, *12*, 74–81; *J. Phys. Chem. C* **2018**, *122*, 5967–5977; *Small* **2022**, *18*, 2202301; *J. Phys. Chem. C* **2023**, *127*, 5783–5790). Therefore, it is possible that in our case the deiodination is already initiated upon RT deposition, with a reaction yield of 20%.

Action: We have now included the statistical analysis of deiodinated species in our revised manuscript, and added the citations to the relevant literature for C-I dissociation temperature on Au(111). For completeness, we have modified the Supplementary Fig. 3 and added an inset that shows a zoom-in of a molecular island, clarifying that the molecules contained therein are intact.

Fig. R1. STM image acquired after deposition of **1** on Au(111) held at RT. Two types of molecular self-assembly are observed, namely SA1 and SA2. Molecular schemes are overlaid to some areas as a guide to the eye. Red circles indicate additional features in SA2, presumably chemisorbed iodine atoms. Scanning parameters: $I_t = 20$ pA, $V_b = -1.0$ V. Acquisition temperature: 4.7 K.

2. *Barrier height/energy barrier is attributed to the energy required for the initial states to convert to transition state along the minimum energy pathway. Many plausible pathways can be calculated by varying different factors, for example bond length/distance, rotation angle, etc, but not all the structures with the highest energy for different plausible pathways are referred to as transition states, but only the one which is along the minimum energy path. The energy of the transition state is associated with one structure, namely the saddle point, along the minimum energy path.*

In the manuscript, the current theoretical data refer to several energies as ‘reaction barrier’ or ‘energy barrier’. Co-authors justify that the presented path is not the minimum energy path, nor the presented mechanism is the only favourable mechanism, but ‘a qualitative overview of the reaction mechanism’, or ‘a qualitative overview of a possible reaction mechanism, and not to pinpoint accurate energy values of states and barriers, which is out of the scope of the present study and better suited for a dedicated computational study.’

If NEB simulation or String method has been considered to be expensive within the resources available to the co-authors, I would suggest that the minimum energy path should be retrieved from calculating potential energy surfaces, at single energy points of molecular structure considering sufficient variables to be able to assign the saddle point, hence, retrieving the energy of the transition state without going through expensive simulation. Otherwise, the current calculations merely show random paths which the molecules might (or not) have taken.

Except the calculated structures which correspond to the experimental data – structures formed after the sequential annealing steps – the context of the theoretical data presented in this manuscript is imprecise (compared to the acceptable methods in literature) and misleading to the readers of Nature Communication journal.

Our reply: We are grateful to the reviewer for this comment, which highlights that our current description of the method used to compute the energy barriers was lacking clarity. Our approach is correct, and we now discuss it widely in the Supplementary Information file (see below). Nevertheless, we have now computed the improved tangent NEBs with the climbing image method for all steps starting from our guess paths. This decision mainly originated from the coincidence pointed out by the reviewer where we observed nearly identical barriers for two de-hydrogenation steps (1.291 eV and 1.292 eV) that now changed to 1.21 eV and 1.26 eV, respectively. In the SI we point out that, if the initial guess for the NEB is accurately determined, in some cases it could be sufficient to use this result and skip the NEB refinement step.

The energy barriers reported in this work are obtained using the climbing image nudged elastic band (CI-NEB) method, from initial guesses derived from series of constrained geometry optimizations (CGO), as described below.

In order to characterize the reactions occurring on the surface, as a first step we identified metastable intermediate states matching the experimentally observed ones. From a starting geometry, we established a chemically valid reaction path using a collective variable (CV), such as an interatomic distance. We slowly adjusted this variable's value (incrementing by 0.05 Å on interatomic distances), consistently optimizing the system's geometry while keeping the CV constrained to its actual value until reaching a new minimum. During this process, we allowed all other degrees of freedom to relax until reaching a minimum energy state. This iterative procedure continued until a new local energy minimum was attained. In each optimization step, all atomic positions were relaxed until the forces acting on the atoms were below 10^{-4} atomic units. After each CGO series, the resulting "end geometry" was re-optimized to identify the most stable local minimum, which served as the starting point for subsequent steps. The highest point along this path approximated a transition state. To refine our estimation of the transition state's energy, we performed CI-NEB calculations (with a convergence threshold on the barrier set at 0.02 eV) on the initial minimum energy paths (MEPs) obtained from CGOs. We utilized 14 replicas for each NEB calculation.

Next, we address the comparison between energy barriers determined from CGOs and CI-NEB calculations. Within standard transition state theory, a series of CGOs results into a MEP, and for each value of the CV there is a well-defined value for the energy of the system. Moreover, the maximum in between two minima along this path is, by definition, a transition state (see e.g. F. Jensen "Introduction to Computational Chemistry" Wiley (1999)). It's important to note that the method of slowly varying a constraint is common in molecular dynamics simulations aimed at performing thermodynamic integration, a method in use for decades that refers back to the coupling parameter method of Kirkwood (J. Chem. Phys. 3, 300 (1935), see e.g. "Understanding Molecular Simulation From Algorithms to Applications" D. Frenkel, B. Smit, Elsevier (1996)). In our scenario, due to discretization in a zero-temperature investigation, achieving the exact transition state becomes improbable, resulting in either 'overshooting'

or 'undershooting' it. This is why, if higher accuracy must be met, it could be necessary to refine the identified MEP with a method such as NEB. However, for a qualitative understanding of a chemical reaction, the computationally intensive NEB calculation might be bypassed, and the energy landscape derived from CGOs could suffice. To support this assertion, we report in the new Supplementary Fig. 9 the energy barriers presented in the main text – obtained via the NEB method – along with those obtained from the series of CGOs. The observed differences between values obtained from these methods consistently remain below 0.08 eV.

As final general comments, we would also like to add the following. Modeling the entire pathway between two experimentally observed states is often impractical. In cases like the one presented, identifying intermediate steps starting from reasonable assumptions about the reaction becomes necessary. For example, transitioning from III to IV (refer to Figure 6 in the manuscript) cannot be computed as a single comprehensive model; it requires breaking down into individual dehydrogenation events. The pathway depicted in Figure 6 results from informed guesses based on chemical intuition rather than an all-encompassing model.

Action: We updated the energy diagram in the main manuscript (Fig. 6) with the barriers obtained from our new NEB calculations and we added in the SI a detailed description of the computational approach.

3. Line 280-305 (Page 9) refer to the C-C bond breaking (SI Fig. 10) and that the energy barriers of 3 eV and 2.2 eV have been calculated in the absence and presence of gold ad-atom, respectively.

As it has been explained in the manuscript, the 2.2 energy is retrieved from the calculation where gold ad-atom is involved in the bond breakage. I assume the calculation of C-C bond breaking with 3 eV energy height has been carried out for the molecule adsorbed on the surface. I would suggest that for this calculation, one gold ad-atom should be similarly located on the surface but in further proximity to the molecule.

Our reply: We thank the reviewer for this comment. In the last sentence of our Methods section, it is stated: "We highlight that a gold adatom was also included in each of the previous computed geometries, but was placed far from the molecular units. This allows a direct comparison of the energies in all steps". We hereby confirm that all the calculations in our work are performed with the molecules adsorbed on the Au(111) substrate (i.e. no gas phase calculations). In both cases of C-C bond breaking with barrier of 3 eV and 2.2 eV a gold adatom was present on the surface. In the former case, it was placed far from the molecule, and in the latter case it was placed next to the reacting site (see Supplementary Fig. 10 in our SI file).

Action: We have modified the corresponding sentence in the Methods section to improve its clarity.

4. In the Page 9, it is written, 'Polymer 4 is very stable and experimentally observed over a wide temperature range'. Then a few lines later, it is stated that, 'The calculated energy barrier of 2.2 eV with adatoms is high enough to justify the stability of polymer 4 below 380 °C, which ensures an extremely high selectivity between two different cyclized products.'

Fig. 6 shows that structure IV has been stabilized by the presence of gold ad-atom (from -2.89 eV to -3.03 eV), which has been converted to the structure V with -4.13 eV. What are the two cyclized products referred here and the attribution to selectivity? For selectivity, there should be at least two competitive pathways to be compared with, hence, concluding one pathway leading to one product is more selective than the other one (starting from the same initial state). This is supported by having the minimum energy path for both, and comparing their barrier height.

For selectivity, do the co-authors mean a comparison between 5-exo and 6-endo?

Our reply: We are grateful to the reviewer for this comment, and apologize if we did not use the correct terminology, which caused confusion. In the indicated section, we referred to the stability of polymer **4** over a wide temperature range. The barrier of 2.2 eV found for the pentagon-to-hexagon rearrangement ensures that high (thermal) energy has to be provided to the system to reach its final stage (polymer **5**), which justifies the stability of polymer **4** and the absence of any modification to its structure from 215 °C to 380 °C. Therefore, if one aims at using vinyl groups to achieve five- or six-membered rings at the armchair edge of polyphenylenes, it is relatively easy to obtain the targeted ring topology with extremely high yield by adjusting the annealing temperature. Indeed, the reviewer is right in pointing out that selectivity is not the right term to be used, and we are not referring to the 5-exo and 6-endo comparison here.

Action: We have rephrased the inaccurate sentence as follows: "It is thus relatively easy to obtain the targeted ring topology – 5- or 6-membered rings as in polymer **4** and **5** – with extremely high yield by adjusting the annealing temperature."

In the Conclusion section, Page 10, it has been stated that structures 4 and 5 are the kinetically and thermodynamically favoured products. In Fig. 6, I don't see any competitive path to the formation of structure 4. Thus, why structure 4 is concluded to be a kinetically favoured product, compared to what other structure?

Please clarify.

Our reply: We would like to thank the reviewer for pointing out this incomplete statement, which could cause confusion. In the conclusions, we state that we can "[...] obtain clean conversion from one architecture to the next with very high yield by annealing the sample to the corresponding temperature". In particular, polymer **4** and **5** are separated by a high barrier of 2.2 eV, and we wanted to further emphasize the stability of the former, although the latter remains the thermodynamically favored product. However, the reviewer is correct and we used an imprecise terminology.

Action: To avoid confusion and repetitions, we have deleted the inaccurate sentence pointed out by the reviewer.

5. I also noticed that there are several 'energy heights' with 1.3 eV. I am curious if co-authors can justify it, as it may suggest a systematic repetition of a factor involved.

Our reply: The reviewer is right in pointing out that there are several energy barriers amounting to 1.3 eV. These are numbers rounded to the first decimal digit, and of course are not identical. Indeed, the barrier in the 6-endo part amounts to 1.329 eV, the one of the first dehydrogenation (with transition state at -0.75 eV) is equal to 1.292 eV, and the barrier of the third dehydrogenation (with transition state at 0.89 eV) is 1.291 eV. After performing the new NEB calculations, we have now reported the energy barriers originating from them. We stress here that there are two steps (grey color in Fig. 6) where we did not compute the energy barriers. Here, we assumed that they are identical to the previous steps, where the same dehydrogenation process occurs at the opposite side of the molecular unit.

Action: The energy landscape in Fig. 6 has been modified in our revised manuscript to better clarify the computational procedure, energy corrections, and assumptions. Moreover, the values of the energy barriers have now been updated with those obtained via the NEB method.

Reviewers' Comments:

Reviewer #1:

Remarks to the Author:

The authors have found an excellent solution to address my remaining comment. Publication of this outstanding manuscript is highly recommended.

Reviewer #3:

Remarks to the Author:

The current manuscript is the resubmission of the study carried out by Prof. Fasel and co-workers, titled "On-Surface Cyclization of Vinyl Groups on Poly-para-phenylene Involving an Unusual Pentagon to Hexagon Transformation".

The manuscript has improved significantly and most of my comments have been addressed in the current version. Congratulations to the co-authors!

The only part which still lacks clarity is the description of the energies given in Figure 6. I have some suggestions, below, but would leave it to the editor and the co-authors to implement the suggestions.

1. In the following statement in page 9, "Moreover, to create a C-C bond in the case of pentagon formation, the reaction undergoes a transition state that is more favored (by 0.32 eV) than in the hexagon formation (Fig. 6).", from which part of the plot the 0.32 eV can be retrieved? The activation energy for the 5-exo path is 1.41 eV, and that of the 6-endo is 0.42 eV + 1.38 eV (for the two steps). The difference between the overall activation energy of the two paths is 0.39 eV. So, where does the 0.32 eV refer to?

2. In the following statement in page 9, "According to the DFT calculations, the activation energy for the removal of hydrogen attached to the apex of the five-membered ring and from the methyl group of III is 1.3 eV and 1.2 eV, respectively (see Fig. 6).", it is not clear from where the 1.3 eV and 1.2 eV have been retrieved.

I suspect the authors have rounded the 1.26 eV and 1.21 eV in the plots. If this is the case, I would suggest keeping two digits and be consistent in reporting all the numbers associated with Figure 6.

3. The figure description of Figure 6 lacks clarity. I would suggest rewriting this part for a better communicating of your results with general readers of the journal.

Also, if the grey-labeled energy barriers were not calculated, then you could simply omit that part from the plot, and instead, only show the energy of the structure without the assumed energy barrier. This is important because the energy barrier might not be exactly the same for two repeating dehydrogenation steps.

4. In the NEB-CI and CGO simulations, please specify in the main manuscript which atoms and /or coordinates have been kept frozen or relaxed, as well as convergence criteria (energy and force). This is important as the proposed mechanism is likely affected by the convergence criteria and the relaxed coordinates.

5. Some editorial points for enhancing the clarity of the manuscript.

In the following statement in the abstract, "Here, we study the reactions of vinyl groups on poly para-phenylene and provide a comprehensive description of all the reaction steps taking place on the Au(111) surface under ultrahigh vacuum conditions."

- vinyl groups are not on the poly para-phenylene, but derived from phenyl groups.
- instead of "all the reaction steps", I would suggest using the proposed steps, or something like this.